# DEGRADATION-AWARE ALL-IN-ONE IMAGE RESTORATION VIA LATENT PRIOR ENCODING

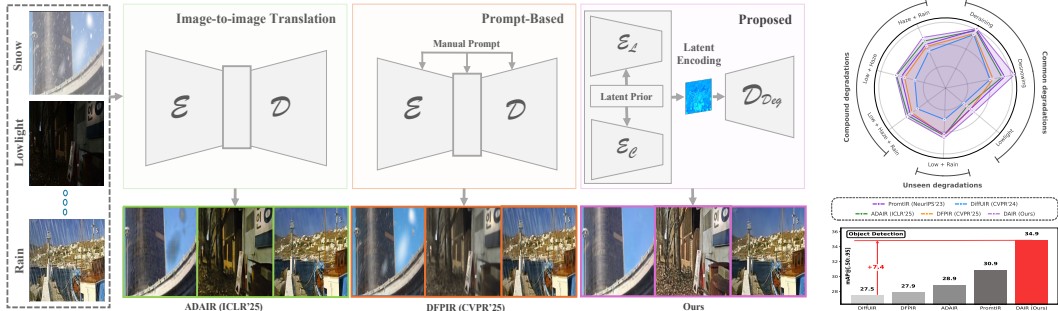

Figure 1: Comparison of common all-in-one image restoration paradigms. Existing approaches depend on explicit task-specific guidance through manual prompts or predefined architectural biases. Our DAIR learns degradation-aware representations directly from degraded images through latent prior inference.

## ABSTRACT

Real-world images often suffer from spatially diverse degradations such as haze, rain, snow, and low-light, significantly impacting visual quality and downstream vision tasks. Existing all-in-one restoration (AIR) approaches either depend on external text prompts or embed hand-crafted architectural priors (e.g., frequency heuristics); both impose discrete, brittle assumptions that weaken generalization to unseen or mixed degradations. To address this limitation, we propose to reframe AIR as learned latent prior inference, where degradation-aware representations are automatically inferred from the input without explicit task cues. Based on latent priors, we formulate AIR as a structured reasoning paradigm: (1) which features to route (adaptive feature selection), (2) where to restore (spatial localization), and (3) what to restore (degradation semantics). We design a lightweight decoding module that efficiently leverages these latent encoded cues for spatially-adaptive restoration. Extensive experiments across six common degradation tasks, five compound settings, and previously unseen degradations demonstrate that our method outperforms state-of-the-art (SOTA) approaches, achieving an average PSNR improvement of 1.68 dB while being three times more efficient. Code will be released upon publication.

## 1 INTRODUCTION

Real-world images are frequently degraded by factors like haze, rain, snow, low-light, and motion blur (Tian et al., 2025a; Jiang et al., 2025). These spatially varying degradations may occur alternately or even overlap, depending on environmental and capture conditions (Guo et al., 2024). Such entangled degradation patterns significantly reduce visual quality, resulting in impaired downstream vision tasks (Tian et al., 2025a; Jiang et al., 2025). Recent deep learning-based single-task methods (Dong et al., 2020; Valanarasu et al., 2022; Chen et al., 2021; Cai et al., 2023; Wang et al., 2022; Zhang et al., 2017) have made significant progress in addressing individual restoration challenges. However, deploying separate task-specific (TS) networks for each degradation type is computationally expensive and impractical, driving interest in AIR frameworks (Jiang et al., 2025).

Current AIR approaches can be broadly categorized into two paradigms. (1) Adaptive feature learning methods, such as PromptIR (Vaishnav et al., 2023), ADAIR (Cui et al., 2024) and AirNet (Li et al., 2022), utilize hand-crafted frequency priors or architectural inductive biases to automatically differentiate degradation types. Although these methods circumvent manual specification, they inherently depend on pre-defined assumptions about degradation characteristics, which often fail to generalize to novel or compound corruptions (Gao et al., 2024). (2) Prompt-based restoration methods, such as UniRestore (Chen et al., 2025), InstructIR (Conde et al., 2024) and DFPIR (Tian et al., 2025a), rely on explicit, manually provided prompts to guide the restoration network by specifying degradation types. These methods offer flexible control but face a fundamental "chicken-and-egg" dilemma: in real-world scenarios, the degradation type and location are rarely known beforehand, yet the network requires this information to perform effective restoration (Jiang et al., 2025). Moreover, TS instructions constrain generalization, particularly in cases of mixed degradations or varying homogeneous degradations (e.g., different noise levels). As illustrated in Fig. 1, both existing AIR paradigms have limited robustness and practical applicability in unconstrained environments.

To overcome these limitations, we reframe AIR as a latent prior inference problem. Unlike existing methods (Chen et al., 2025; Conde et al., 2024; Tian et al., 2025a) that rely on explicit degradation prompting or predefined architectural inductive biases, we propose learning degradation-aware representations directly from the degraded image through a multi-level feature descriptor inspired by the variational autoencoder (VAE) (Kingma & Welling, 2013). Our learned prior eliminates the need for manual degradation hints and enables spatially adaptive restoration of diverse and unseen degradations. Guided by these learned priors, the proposed degradation-aware AIR (DAIR) framework incorporates a "where, which, what" reasoning paradigm: it learns degradation-aware feature selection (which), localizes corrupted regions using spatial attention maps (where), and adaptively fuses multi-scale global representations (what). Our unified strategy substantially enhances the flexibility and generalizability of blind restoration methods.

**Our contributions are:**

– We propose reframe AIR to directly learn latent degradation priors from the corrupted image, eliminating the need for external manual prompts (MP).

– We propose a reasoning image restoration paradigm comprising: (1) latent priors for learning degradation-aware representations, enabling decisions on "which" encoder features to utilize for reconstruction; (2) spatially-adaptive degradation map (DM) that integrate frequency-domain cues with efficient element-wise attention, providing interpretable and localized restoration guidance on "where" to focus beyond implicit attention; (3) cross-modal fusion of structural and color cues with global degradation priors via adaptive scaling and shifting to determine "what" content to reconstruct; and (4) a decoder with linear complexity performing explicit spatial reasoning for leverages 1-3 cues, we termed it 3WD (which–where–what decoding).

– DAIR consistently outperforms SOTA on six common restoration tasks (e.g., snow, low-light), five compound degradations (e.g., haze + rain, low-light + haze + rain), and unseen degradations, achieving an average 1.68 dB PSNR gain. It also improves downstream tasks on images with unknown multi-type degradations, e.g., boosting YOLOv12-L (Tian et al., 2025b) object detection (OD) by up to 7.40 mAP over SOTA AIR methods, highlighting strong generalizability (Fig. 1).

## 2 RELATED WORKS

### 2.1 IMAGE-TO-IMAGE TRANSLATION

Early image restoration methods were designed to address individual TS settings, such as denoising (Zhang et al., 2017; Pang et al., 2021; Zhang et al., 2018a), dehazing (Dong et al., 2020; Qin et al., 2020), deraining(Jiang et al., 2020; Ren et al., 2019), low-light image enhancement (LLIE) (Wei et al., 2018; Yi et al., 2023), or deblurring (Cho et al., 2021; Nah et al., 2017). These task-dependent approaches achieved impressive results; their limited scope prevented generalization across diverse restoration challenges. Transformer-based models, such as Restormer (Zamir et al., 2022) and Uformer (Wang et al., 2022), have explored multiple degradation scenarios; however, they still require separate training for individual tasks. To address this, recent methods shift towards AIR, aiming to handle diverse degradations within a unified framework. AirNet (Li et al., 2022) pio-

neered this field by introducing contrastive learning to extract degradation representations, guiding restoration for unknown corruptions. ADAIR (Cui et al., 2024) recalibrate features using adaptive frequency statistics, enabling task-agnostic restoration. Recent methods, such as DiffUIR (Zheng et al., 2024), have also explored generative techniques (i.e., latent diffusion (Rombach et al., 2022)). These unified approaches mark a significant step forward, minimizing TS training and advancing toward generalizable AIR solutions.

## 2.2 PROMPT-GUIDED ALL-IN-ONE RESTORATION

Image-to-image translation-based method commonly fails in separating unique degradations (Brooks et al., 2023; Jiang et al., 2025). To counter this, recent AIR (Gao et al., 2024; Conde et al., 2024; Chen et al., 2025; Tian et al., 2025a) methods have incorporated MP to provide prior knowledge of degradation cues. InstructIR (Conde et al., 2024) and UniRestore (Chen et al., 2025) integrates text-based prompts into a transformer framework, leveraging degradation-specific semantic information to guide the restoration process. OneRestore (Guo et al., 2024) introduced visual and text prompts within a transformer-based framework to encode TS information. Recently, DF-PIR (Tian et al., 2025a) introduced a degradation-aware feature perturbation method that utilizes CLIP-encoded (Radford et al., 2021) text prompts to guide channel shuffling and attention masking, enabling unified restoration across diverse degradations. UHD-Processor (Liu et al., 2025) introduced a VAE-based framework with progressive frequency learning and MP for ultra-high-definition image restoration. Notably, unlike UHD-Processor and other prompt-based methods, our method learns latent degradation representations directly from the input, enabling blind restoration without user guidance.

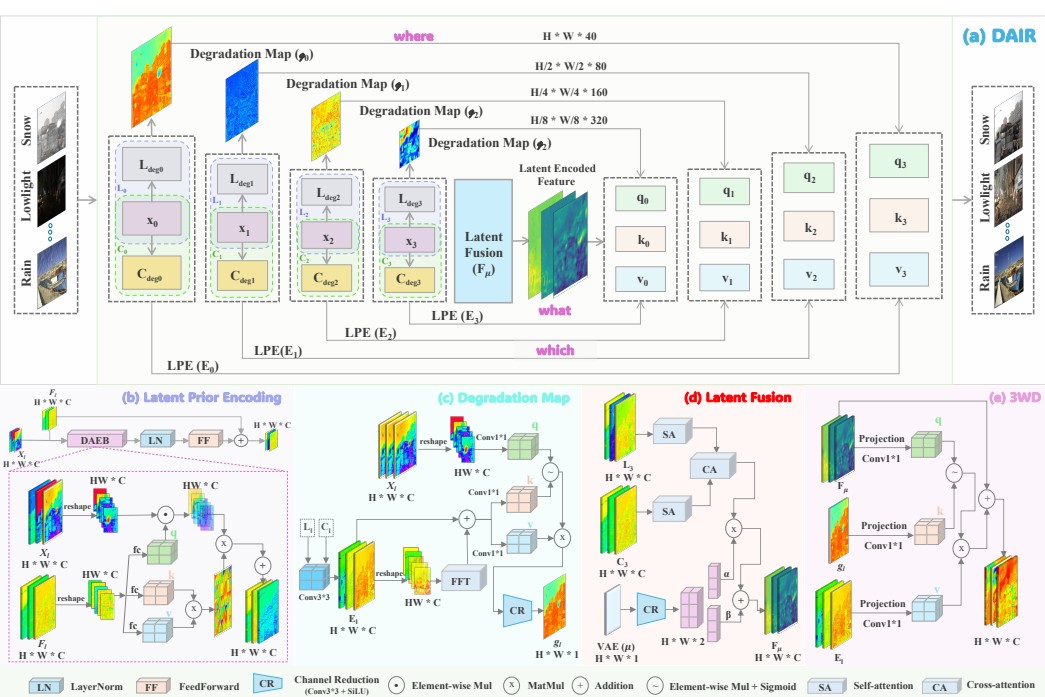

Figure 2: Overview of the proposed DAIR framework. (a) The overall architecture consists of an encoder-decoder structure complemented by degradation priors from a VAE encoder. The degraded image is encoded by our Latent Prior Encoding (LPE) in both luminance and chrominance spaces. (b) Latent Prior Encoding (*which*) comprises multiple Degradation Aware Encoder Blocks (DAEB) that encode degradation prior information from the VAE. Latents encoded at different stages are passed to the Degradation Map blocks. (c) The Degradation Mapping (*where*) block utilizes LPE latents and VAE priors to generate degradation maps for each stage. (d) The Latent Fusion (*what*) block combines luminance and chrominance latents to generate unified latents.

## 3 METHOD

**Overall pipeline.** Fig. 2 illustrates the details of our framework. Given a degraded input image $\mathbf{I}_{deg} \in \mathbb{R}^{3 \times H \times W}$ with unknown degradation $\mathcal{D}_R$, our framework learn to infer stage-wise latent codes $x\ell$ ($\ell = 0, \ldots, 3$) and a bottleneck global descriptor $\mu$, serving as degradation priors. The encoder employs a two-branch architecture that separately processes multi-scale structural features (e.g., luminance $L_\ell$) and color features (e.g., chrominance $C_\ell$), enabling more effective representation of both spatial and chromatic information (Sharif et al., 2025; Yan et al., 2025). At each stage, $L_\ell$ and $C_\ell$ interact with $x_\ell$ via *Degradation-Aware Encoder Blocks* (DAEB), leveraging degradation information into the feature extraction process. The branches are fused as $F_\ell = L_\ell + C_\ell$ and used as residual refined feature propagation to the decoder. Concurrently, a mapping block transforms $(L_\ell, C_\ell, x_\ell)$ into a compact DM $g_\ell$, which is also forwarded to the decoder for explicit degradation localization guidance. At the bottleneck, a $\mu$-guided fusion block merges the deepest LC features with the global degradation latent using adaptive scaling and shifting (Perez et al., 2018), resulting in $F_\mu = \gamma \odot [L_3 + C_3] + \beta$. The decoder reconstructs the clean image from $F_\mu$, leveraging the degradation focus map $g_\ell$ and the selected features $F_\ell$ derived from latent priors. This architecture enables the model to automatically identify and restore a wide range of degradations without explicit user input or task specification.

### 3.1 LEARNING DEGRADATION REASONING WITH LATENT PRIOR

We propose a multi-scale latent descriptor (Kingma & Welling, 2013) that directly infers continuous degradation representations from corrupted images without user-guided MP. Given $\mathbf{I}_{deg} \in \mathbb{R}^{3 \times H \times W}$, the encoder extracts hierarchical feature maps $\{\mathbf{x}_\ell\}_{\ell=0}^3$ using convolutional layers and residual channel-attention (Hu et al., 2018) blocks that emphasize degradation-relevant cues. At the deepest stage, multi-head self-attention (Vaswani et al., 2017) refines the features to obtain the bottleneck representation $\mathbf{z}$. The VAE parameterizes a distribution with $\boldsymbol{\mu} = f_\mu(\mathbf{z})$ and $\log \boldsymbol{\sigma}^2 = f_\sigma(\mathbf{z})$, sampling $\mathbf{z}_{reparam} = \boldsymbol{\mu} + \boldsymbol{\sigma} \odot \boldsymbol{\epsilon}$ where $\boldsymbol{\epsilon} \sim \mathcal{N}(0, \mathbf{I})$. The decoder reconstructs the input image from the sampled latent. This multi-objective training ensures that hierarchical features $\{\mathbf{x}_\ell\}$ capture degradation-specific representations at multiple scales, while bottleneck statistics $\boldsymbol{\mu}$ encode global degradation semantics. Both serve as degradation-aware latent cues to condition the restoration model:

$$\text{VAE}(\mathbf{I}_{deg}) = \{\mathbf{x}_\ell\}_{\ell=0}^3, \ \boldsymbol{\mu}. \tag{1}$$

### 3.1.1 LATENT PRIOR ENCODING

Unlike existing methods (DFPIR (Tian et al., 2025a), CAPTNet (Gao et al., 2024)) that inject prompts during decoding, we inject VAE-derived degradation priors during encoding. This enables our encoders to generate both degradation-aware and degradation-agnostic skip connections, providing richer guidance for reconstruction. We separately encode the luminance and chrominance (LC) components (Smith, 1978)($\mathbf{L}_{deg}, \mathbf{C}_{deg}$) of the degraded input using dedicated LC encoders (Yan et al., 2025; Sharif et al., 2025) for better structural and chromatic information. Fig. 2 (b) illustrates our latent prior encoding. Two parallel encoders, $\mathcal{E}_L(\cdot)$ and $\mathcal{E}_C(\cdot)$, process these components across multiple scales using DAEB. At stage $\ell$, we injects latent degradation priors $\mathbf{x}_\ell$ via value modulation:

$$Q_\ell = \mathbf{f}_\ell W_Q, \quad K_\ell = \mathbf{f}_\ell W_K, \quad \tilde{V}_\ell = (\mathbf{f}_\ell W_V) \odot \mathbf{x}_\ell \tag{2}$$

where $\mathbf{f}_\ell \in \{\mathbf{f}_\ell^L, \mathbf{f}_\ell^C\}$ are intermediate features. The attention mechanism produces degradation-aware representations as $\text{DAEB}(\mathbf{f}_\ell, \mathbf{x}_\ell) = \text{softmax}(Q_\ell K_\ell^\top / \sqrt{d})\tilde{V}_\ell$. Combining these components, the encoder outputs at each stage are:

$$L_\ell = \mathcal{E}_\ell^L(\mathbf{L}_{deg}, \mathbf{x}_\ell), \quad C_\ell = \mathcal{E}_\ell^C(\mathbf{C}_{deg}, \mathbf{x}_\ell) \tag{3}$$

At each stage, latent-derived priors $\mathbf{x}_\ell$ modulate the value features after multi-head projection, enabling semantically refined propagation that preserves both degradation-specific and clean structural information. During decoding, these priors serve as selective feature utilization for effective reconstruction.

### 3.1.2 LEARNABLE DEGRADATION MAP

Traditional skip connections propagate noise from degraded inputs to the decoder, without spatial awareness of the degradation (Mao et al., 2016). We counter these limitations with learnable DM that encode spatially-varying degradation patterns for effective reconstruction (Fig. 2 (c)). At encoder stage $\ell$, our *Degradation Mapping Block* generates scale-specific maps $g_\ell$ from LC features $L_\ell, C_\ell$ and latent prior encoding $\mathbf{x}_\ell$. We first fuse complementary LC information: $\mathbf{f}_\ell^{LC} = \text{Conv}_{3\times3}(L_\ell + C_\ell)$. To capture frequency-domain degradation characteristics, we enhance features with spectral information via Fast Fourier Transform (Cooley & Tukey, 1965): $\mathcal{F}(L_\ell) = \text{FFT}(L_\ell - \mu(L_\ell))$, $\mathcal{F}(C_\ell) = \text{FFT}(C_\ell - \mu(C_\ell))$. The magnitude and phase components are concatenated and projected:

$$\mathbf{f}_\ell^{\text{freq}} = \text{ReLU}(\text{Conv}_{1\times1}([|\mathcal{F}(L_\ell)| \,\|\, |\mathcal{F}(C_\ell)| \,\|\, \angle\mathcal{F}(L_\ell) \,\|\, \angle\mathcal{F}(C_\ell)])), \quad \tilde{\mathbf{f}}_\ell^{LC} = \mathbf{f}_\ell^{LC} + \mathbf{f}_\ell^{\text{freq}} \quad (4)$$

For computational efficiency, we employ element-wise attention:

$$Q_\ell = \text{Conv}_{1\times1}(\tilde{\mathbf{f}}_\ell^{LC}), \quad K_\ell = V_\ell = \text{Conv}_{1\times1}(\mathbf{x}_\ell) \quad (5)$$

$$\mathbf{A}_\ell = \sigma(Q_\ell \odot K_\ell), \quad \mathbf{y}_\ell = \mathbf{A}_\ell \odot V_\ell \quad (6)$$

where $\sigma$ is sigmoid activation and $\odot$ denotes Hadamard product. The DM is generated via residual refinement:

$$g_\ell = \phi_{\text{map}}(\text{ReLU}(\text{Conv}_{3\times3}(\mathbf{y}_\ell + \tilde{\mathbf{f}}_\ell^{LC}))) \quad (7)$$

where $\phi_{\text{map}}$ is a two-layer convolutional head producing $g_\ell \in \mathbb{R}^{1 \times H_\ell \times W_\ell}$. This design captures spatial-frequency degradation patterns with linear complexity, while providing interpretable and localized restoration guidance.

### 3.1.3 LATENT ENCODED FUSION

At the encoder bottleneck, we leverage the deepest LC features $\mathbf{L}_3, \mathbf{C}_3 \in \mathbb{R}^{B \times 320 \times \frac{H}{8} \times \frac{W}{8}}$, along with the latent global descriptor $\mu$. As shown in Fig. 2 (d), our fusion mechanism integrates three components: (i) intra-branch self-attention (Sharif et al., 2025; Cai et al., 2023) for independent modality refinement, where $\hat{\mathbf{L}}_3 = \text{MHSA}(\mathbf{L}_3)$ and $\hat{\mathbf{C}}_3 = \text{MHSA}(\mathbf{C}_3)$; (ii) cross-branch attention (Sharif et al., 2025) for complementary information exchange, where chrominance attends to luminance: $\mathbf{F} = \text{LayerNorm}\left(\text{MHCA}(\hat{\mathbf{C}}_3, \hat{\mathbf{L}}_3) + \hat{\mathbf{C}}_3 + \hat{\mathbf{L}}_3\right)$, enabling the model to leverage structural and color information while preserving individual branch characteristics; and (iii) adaptive feature modulation conditioned on the global degradation prior $\mu$, combining LC features through adaptive scaling and shifting for TS feature modulation while maintaining underlying image characteristics. Unlike standard FiLM (Perez et al., 2018) that applies uniform modulation, our approach makes degradation-aware reconstruction:

$$[\boldsymbol{\gamma}_{\text{struct}}, \boldsymbol{\beta}_{\text{color}}] = \phi_\mu(\mu), \quad \boldsymbol{\gamma}_{\text{struct}}, \boldsymbol{\beta}_{\text{color}} \in \mathbb{R}^{B \times 320 \times \frac{H}{8} \times \frac{W}{8}} \quad (8)$$

$$\mathbf{F}_\mu = \mathbf{F}_{\text{LC}} \odot (1 + \boldsymbol{\gamma}_{\text{struct}}) + \boldsymbol{\beta}_{\text{color}} \quad (9)$$

where $\phi_\mu$ serves as a content decision network (two $1 \times 1$ convolutions with SiLU) that determines what structural and chromatic content should be reconstructed based on inferred degradation characteristics. Our **identity-anchored scaling** $(1 + \boldsymbol{\gamma}_{\text{struct}})$ ensures content reconstruction decisions are made as learned adjustments around the original cross-modal features, completing our "what" reasoning component.

## 3.2 DEGRADATION-AWARE RECONSTRUCTION

The proposed decoder addresses our fundamental reasoning questions by reconstructing images through the *3WD* module (Fig. 2 (e)). At each decoder stage $\ell$, given upsampled features $U_{\ell-1}$ and combined encoder features $E_\ell = L_\ell + C_\ell$, we compute linear attention projections:

$$Q_\ell = W_Q^{(\ell)} U_{\ell-1}, \quad K_\ell = W_K^{(\ell)} g_\ell, \quad V_\ell = W_V^{(\ell)} E_\ell$$

Degradation-guided attention operates as $A_\ell = \sigma(Q_\ell \odot K_\ell)$, enabling $g_\ell$ to directly modulate spatial restoration. Features are updated through:

$$D_\ell = \phi_\ell(A_\ell \odot V_\ell + U_{\ell-1}) \tag{10}$$

Notably, our proposed 3WD achieves linear computational complexity of $O(HW)$ per stage, in contrast to the quadratic complexity $O(H^2W^2)$ of standard attention mechanisms (Vaswani et al., 2017). This efficiency stems from utilizing element-wise multiplication ($\odot$) between tensors of shape $H \times W \times C$, requiring exactly $HW \times C$ operations. We perceived final reconstruction with learned features and global residual: $\hat{\mathbf{I}} = \tanh(W_{\text{rec}} D_1) + \mathbf{I}_{\text{deg}}$.

## 4 EXPERIMENTS

### 4.1 SETUP

**Dataset and methods.** We evaluated our method under two degradation scenarios: common (non-overlapping) degradations and compound (overlapping) settings. For common degradations, we combined six widely used tasks, including dehazing (SOTS (Li et al., 2018)), deraining with heavy rain (Rain100H (Fu et al., 2017)), desnowing (CCD (Li et al., 2020)), real-world deblurring (Nah et al., 2017), deblurring (GoPro (Nah et al., 2017)), denoising (DIV2K (Agustsson & Timofte, 2017)) with random noise levels ($\sigma \in [0, 50]$) to improve generalization, and real-world LLIE using the LSD dataset (Sharif et al., 2025). In compound degradation, we employed the CDD dataset (Guo et al., 2024), combining five degradations (e.g., haze+rain, low-light+haze+snow). Subsets like haze+snow and low-light+rain were reserved for testing generalization comparison on unseen degradations. We also included several real-world out-of-distribution datasets, such as SIDD for noisy images, LSD-U for unseen low-light scenarios, underwater image enhancement datasets, and medical image enhancement and denoising datasets.

We compared DAIR against transformer-based baseline methods (Uformer (Wang et al., 2022), Restormer (Zamir et al., 2022)), SOTA AIR approaches (ADAIR (Cui et al., 2024), AIRNet (Li et al., 2022)), prompting-based methods (DFPIR (Tian et al., 2025a), PromptIR (Vaishnav et al., 2023)), and diffusion-based latent enhancement (DiffUIR (Zheng et al., 2024)). Single-task benchmarks included deraining (HDCWNet (Zhu et al., 2021), TransWeather (Valanarasu et al., 2022)), Retinex-based LLIE (RetinexNet (Wei et al., 2018), Diff-Retinex (Yi et al., 2023)).

**Implementation** We first pre-train our latent encoder for 200,000 steps, combining six single and five compound (known) degradations using a composite loss (self-reconstruction + KL (Kingma & Welling, 2014) + discriminative latent regularizer (Guo et al., 2023)). The resulting encoder is frozen and reused for all single, compound, and unseen degradation experiments without any fine-tuning. We only tune the main network for TS settings using the combination of L1 and SSIM losses for fair comparison. This network has been trained for 100,000 to 500,000 steps, depending on task complexity. We trained both the latent encoder and the main restoration network using the Adam optimizer with hyperparameters $\beta_1 = 0.9$, $\beta_2 = 0.99$, and learning rate = 1e-4. All training was conducted on a single NVIDIA RTX 3060 GPU with a batch size of 4, using randomly cropped $256 \times 256$ patches as input. All baseline methods are trained using their official implementations and suggested hyperparameters to ensure fairness.

### 4.2 RESULTS ON COMMON DEGRADATION

#### 4.2.1 MULTI-TASK (6D) RESTORATION

Table 1 and Figure 3 illustrate DAIR's performance across six common degradations under all-in-one setting. Achieving the highest average **PSNR (28.15)** and **SSIM (0.8829)**, DAIR delivers significant improvements, including **+5.64 PSNR** and **+0.0758 SSIM** in desnowing, and **+1.54 PSNR** and **+0.0481 SSIM** in lowlight enhancement. Despite requiring only **45G FLOPs**, DAIR outperforms computationally intensive methods like ADAIR (Cui et al., 2024)(**147G FLOPs**) while operating without MP, ensuring scalability and adaptability. By leveraging degradation-aware latent priors, DAIR generates robust, high-quality restorations across diverse degradation types, making it ideal for real-world applications like autonomous systems and healthcare imaging.

Table 1: Performance comparison across six different image restoration tasks under all-in-one setting: a unified model is trained on a combined set of images obtained from all degradation types and levels. Best results in **bold red**, second best underlined, and increment over best performing method highlighted in **blue**.

| Method | MP | Params (M) | GFLOPs | Lowlight PSNR↑/SSIM↑/LPIPS↓ | Dehazing PSNR↑/SSIM↑/LPIPS↓ | Denoising PSNR↑/SSIM↑/LPIPS↓ | Desnowing PSNR↑/SSIM↑/LPIPS↓ | Deblurring PSNR↑/SSIM↑/LPIPS↓ | Deraining PSNR↑/SSIM↑/LPIPS↓ | Average PSNR↑/SSIM↑/LPIPS↓ |
|---|---|---|---|---|---|---|---|---|---|---|
| Uformer | ✗ | 20.63 | 43.86 | 12.10/0.4897/0.3621 | 29.15/0.9727/0.0195 | 27.55/0.8201/0.1334 | 24.80/0.8832/0.0904 | 23.67/0.8074/0.2401 | 28.74/0.8823/0.1051 | 24.34/0.8092/0.1584 |
| Restormer | ✗ | 26.13 | 141.75 | 15.33/0.6180/0.2697 | 28.51/0.9738/0.0154 | 20.41/0.5357/0.4148 | 22.98/0.8585/0.1213 | 25.41/0.8286/0.2080 | 25.99/0.8164/0.1547 | 23.10/0.7718/0.1973 |
| AIRNet | ✗ | 8.93 | 301.27 | 8.87/0.3655/0.3679 | 25.92/0.9663/0.0237 | 28.87/0.8789/0.1020 | 19.08/0.7300/0.2596 | 22.81/0.8067/0.2284 | 24.92/0.8027/0.1692 | 21.75/0.7583/0.1918 |
| PromptIR | ✗ | 35.59 | 158.14 | 14.73/0.5374/0.3665 | 30.41/0.9580/0.0221 | 23.96/0.6378/0.2339 | 27.08/0.9169/0.0634 | 26.04/0.8366/0.2069 | 30.01/0.8992/0.0669 | 25.37/0.7977/0.1600 |
| DiffUIR | ✓ | 36.26 | 4400.00 | 10.76/0.4101/0.4290 | 30.05/0.9797/0.0151 | 20.25/0.6859/0.2329 | 20.39/0.8014/0.1709 | 25.03/0.8474/0.2019 | 27.46/0.8917/0.0719 | 22.32/0.7694/0.1870 |
| ADAIR | ✗ | 28.78 | 147.18 | 12.76/0.4974/0.3605 | 30.15/0.9783/0.0135 | 28.65/0.8715/0.1043 | 25.86/0.8812/0.0966 | 25.14/0.8200/0.2307 | 29.56/0.8901/0.0889 | 25.35/0.8231/0.1491 |
| DFPIR | ✓ | 31.07 | 151.07 | 14.84/0.5232/0.3766 | 28.29/0.9709/0.0214 | 27.78/0.8341/0.1433 | 21.74/0.7935/0.2189 | 26.10/0.8271/0.2230 | 28.02/0.8372/0.1506 | 24.46/0.7977/0.1890 |
| **DAIR (Ours)** | ✗ | 18.08 | 45.65 | 16.87/0.6661/0.2559 | 34.08/0.9864/0.0100 | 29.12/0.8971/0.0826 | 31.50/0.9570/0.0264 | 26.82/0.8793/0.1653 | 30.51/0.9117/0.0663 | 28.15/0.8829/0.1011 |
| *Improvement* | | -10.70 | -96.10 | +1.54/+0.0481/-0.0138 | +3.67/+0.0067/-0.0035 | +0.25/+0.0182/-0.0194 | +4.42/+0.0401/-0.0370 | +0.72/+0.0427/-0.0366 | +0.50/+0.0125/-0.0006 | +2.78/+0.0598/-0.0480 |

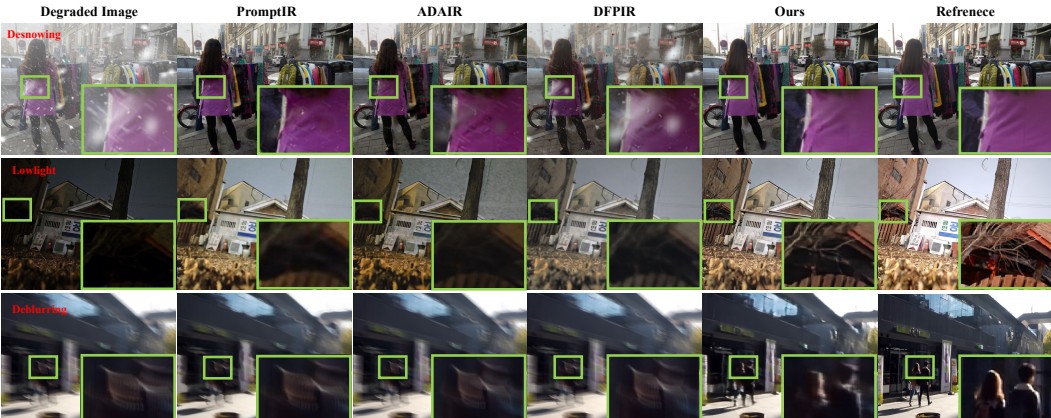

Figure 3: Visual comparison for six common degradation settings under the all-in-one setting. The proposed method produces consistent and plausible images compared to the existing methods.

### 4.2.2 SINGLE-TASK EVALUATION: LLIE AND DESNOWING

We evaluated DAIR on single restoration tasks to demonstrate its adaptability. Table 2 shows DAIR achieves **32.75 PSNR / 0.9632 SSIM** for desnowing, outperforming HDCWNet (Zhu et al., 2021) by **+3.50 PSNR / +0.0308 SSIM**. For LLIE, DAIR achieves **16.30 PSNR / 0.6634 SSIM**, improving upon Restormer (Zamir et al., 2022) by **+0.52 PSNR / +0.0450 SSIM**. By leveraging degradation-aware latent priors, DAIR can also adapt to varying degradation types without external MP.

Table 2: Performance comparison on desnowing and LLIE tasks.

| (a) Desnowing | | (b) Low-light enhancement | |
|---|---|---|---|
| **Method** | PSNR↑/SSIM↑/LPIPS↓ | **Method** | PSNR↑/SSIM↑/LPIPS↓ |
| HDCWNet | 29.25/0.9171/0.0517 | RetinexNet | 14.19/0.5183/0.3812 |
| TransWeather | 23.30/0.7631/0.1739 | Diff-Retinex | 15.38/0.5038/0.3841 |
| Uformer | 28.86/0.9324/0.0904 | Uformer | 13.05/0.5716/0.3621 |
| Restormer | 23.21/0.8686/0.1059 | Restormer | 15.78/0.6184/0.3372 |
| ADAIR | 28.31/0.9300/0.0457 | ADAIR | 15.19/0.6094/0.3325 |
| **DAIR (Ours)** | **32.75/0.9632/0.0213** | **DAIR (Ours)** | **16.30/0.6634/0.3279** |
| *Improvement* | **+3.50/+0.0308/-0.0244** | *Improvement* | **+0.52/+0.0450/-0.0046** |

### 4.3 RESULTS ON COMPOUND DEGRADATION

Table 3: Quantitative evaluation of restoration performance on **five** compound degradation types, derived from combinations of low-light, rain, haze, and snow datasets.

| Method | Haze + Rain PSNR↑/SSIM↑/LPIPS↓ | Low + Haze PSNR↑/SSIM↑/LPIPS↓ | Low + Haze + Rain PSNR↑/SSIM↑/LPIPS↓ | Low + Haze + Snow PSNR↑/SSIM↑/LPIPS↓ | Low + Snow PSNR↑/SSIM↑/LPIPS↓ | Average PSNR↑/SSIM↑/LPIPS↓ |
|---|---|---|---|---|---|---|
| Uformer | 22.49/0.8889/0.0867 | 21.37/0.8252/0.1695 | 20.61/0.7838/0.1908 | 18.64/0.6802/0.3052 | 20.31/0.7150/0.2513 | 20.68/0.7786/0.2007 |
| Restormer | 23.57/0.8825/0.0959 | 21.44/0.7753/0.1884 | 20.53/0.7265/0.2367 | 18.46/0.6494/0.3309 | 19.97/0.6664/0.3003 | 20.79/0.7400/0.2304 |
| AIRNet | 18.58/0.7977/0.1987 | 11.73/0.6905/0.2587 | 13.58/0.6320/0.3731 | 12.87/0.5898/0.4141 | 13.76/0.5759/0.4372 | 14.10/0.6572/0.3364 |
| PromptIR | 21.78/0.8818/0.1012 | 20.63/0.8231/0.1850 | 19.83/0.7705/0.2223 | 18.42/0.6801/0.3171 | 20.74/0.7070/0.2645 | 20.28/0.7725/0.2180 |
| DiffUIR | 18.42/0.8022/0.1758 | 14.28/0.7150/0.2562 | 15.79/0.6568/0.3308 | 15.33/0.6238/0.3500 | 12.60/0.6292/0.2988 | 15.28/0.6854/0.2823 |
| ADAIR | 23.51/0.8966/0.0791 | 22.78/0.8420/0.1164 | 21.06/0.7908/0.1774 | 20.46/0.7229/0.2416 | 21.35/0.7215/0.2444 | 21.83/0.7948/0.1769 |
| DFPIR | 20.81/0.8308/0.1548 | 19.01/0.7784/0.2339 | 18.82/0.7139/0.2800 | 15.58/0.6307/0.3366 | 19.70/0.6402/0.3227 | 18.78/0.7188/0.2656 |
| **DAIR (Ours)** | **25.25/0.9259/0.0637** | **23.15/0.8541/0.1408** | **22.03/0.8200/0.1618** | **20.81/0.7613/0.2233** | **21.79/0.7772/0.1874** | **22.61/0.8277/0.1554** |
| *Improvement* | **+1.68/+0.0293/-0.0154** | **+0.37/+0.0121/-0.0011** | **+0.97/+0.0292/-0.0156** | **+0.35/+0.0384/-0.0183** | **+0.44/+0.0557/-0.0570** | **+0.78/+0.0329/-0.0215** |

Figure 4: Visual comparison for five compound degradations. The proposed DAIR can handle compound degradation and produce visually pleasing images.

We evaluated DAIR on five compound degradation scenarios derived from combinations of low-light, rain, haze, and snow datasets. As shown in Table 3, DAIR achieves the highest average **PSNR of 22.61** and **SSIM of 0.8277**, outperforming the next-best method (ADAIR (Cui et al., 2024)) by **+0.78 PSNR / +0.0329 SSIM**. Notable improvements include "Haze + Rain" (**+1.68 PSNR / +0.0293 SSIM**) and "Low + Snow" (**+0.44 PSNR / +0.0557 SSIM**). DAIR effectively restores overlapping degradations, demonstrating robust performance across complex scenarios. Fig. 4 highlights DAIR's superior visual quality, producing artifact-free, natural images.

## 4.4 LATENT PRIOR GUIDED UNSEEN TASK RESTORATION

Our latent prior descriptor enables robust handling of unknown degradations while maintaining clear separation from known ones. As shown in Table 4, the VAE-based encoder learns degradation characteristics directly from corrupted images, achieving strong clustering for both seen and unseen degradations with KNN accuracy of **0.994/0.976**, separation ratios of **2.911/1.523**, and optimal global metrics (lowest Davies-Bouldin index, highest Calinski-Harabasz score). This learned prior guides the main network with degradation-specific descriptions, enabling clear latent separation for different degradation, where existing methods fail (Fig. 5(b)). Consequently, DAIR restores unknown compound degradations effectively (Fig. 5(a)), achieving average gains of **+1.47 dB PSNR** and **+0.0412 SSIM** over SOTA methods (Table 5). Please refer appendix for more details.

Table 4: Latent prior descriptor yields strong clustering and separation for seen/unseen degradations.

| Model | Seen | | Unseen | | Overall | | Global Metrics | |
|---|---|---|---|---|---|---|---|---|
| | Acc↑ | Sep↑ | Acc↑ | Sep↑ | Acc↑ | Sep↑ | DB↓ | CH↑ |
| w/o SupCon | 0.864 | 2.00 | 0.787 | 1.32 | 0.874 | 1.79 | 2.69 | 254.48 |
| Steps = 0 | 0.826 | 3.11 | 0.756 | 1.33 | 0.844 | 2.28 | 4.84 | 458.34 |
| Steps = 100k | 0.871 | 2.36 | 0.828 | 1.27 | 0.892 | 1.95 | 2.45 | 289.84 |
| **Full Training** | **0.994** | **2.91** | **0.976** | **1.52** | **0.991** | **3.22** | **1.23** | **549.34** |

Table 5: Performance comparison on unknown compound degradations.

| Method | Haze + Snow PSNR↑/SSIM↑ | Low + Rain PSNR↑/SSIM↑ | Average PSNR↑/SSIM↑ |
|---|---|---|---|
| ADAIR | 16.07/0.8130 | 21.67/0.7877 | 18.87/0.8003 |
| PromptIR | 17.84/0.8032 | 21.15/0.7752 | 19.50/0.7892 |
| DFPIR | 14.50/0.7577 | 21.52/0.7021 | 18.01/0.7299 |
| **DAIR (Ours)** | **18.65/0.8531** | **23.29/0.8299** | **20.97/0.8415** |
| *Improvement* | **+0.81/+0.0499** | **+1.62/+0.0422** | **+1.47/+0.0523** |

(a)                                                        (b)

Figure 5: DAIR performance on unseen tasks: (a) Visual results for unseen compound degradation (low-light + rain); (b) t-SNE embeddings showing separation of unseen and known degradations.

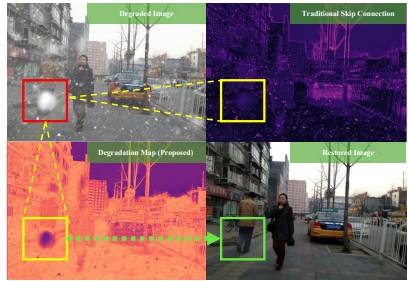

Figure 6: Latent prior encoding validation. (a) Degradation separability via distance matrix. (b) Encoder progression: structural (L1–L2) to chromatic (L3–L4) features.

Figure 7: Degradation localization. Our learnable map precisely identifies affected regions (yellow box).

### 4.5 ANALYSIS OF REASONING RESTORATION

#### 4.5.1 LATENT PRIOR ENCODING

The distance matrix in Fig. 6 (a) shows aggregated latent representations: strong intra-class consistency (diagonal: 160–194) and clear inter-class separation. Related degradations have moderate distances (e.g., Haze–Lowlight: 102, Haze–Blur: 86), while dissimilar pairs show larger separations (e.g., Snow–Lowlight: 201, Snow–Haze: 136). As shown in Fig. 6 (b), encoder stages learn progressive feature selection based on these latent priors: early stages (L1–L2) show higher distances ( 180–200) indicating stronger luminance preference for structural information, while deeper stages (L3–L4) show lower distances ( 60–100) reflecting increased chrominance preference for color-specific restoration. This degradation-aware hierarchical guidance enables DAIR to adaptively leverage structural and chromatic information across different encoder depths for different degradations, validating our "which" reasoning.

#### 4.5.2 DEGRADATION MAP

Table 6 quantitatively evaluates our degradation map localization accuracy ("where"). To evaluate degradation localization, we generate ground truth masks by computing pixel-wise absolute differences $|I_{\text{degraded}} - I_{\text{clean}}|$ and compare them with our predicted degradation maps. Our method achieves a mean IoU of **0.861±0.169** and a Dice score of **0.912±0.118**. Pixel-level degradations (Noise, Lowlight, Haze) achieve near-perfect localization (IoU > 0.95), while spatially-varying degradations (Rain, Snow, Blur) maintain strong performance despite higher variance. As shown in Fig. 7, our learnable degradation maps precisely identify affected regions and significantly outperform traditional skip connections, enabling the decoder to focus reconstruction on degraded areas.

Table 6: Quantitative evaluation of degradation map localization ("where") using IoU and Dice metrics across pixel-wise and spatially-varying degradations.

| Metric | Noise | Lowlight | Haze | Rain | Snow | Blur | Mean ± Std |
|---|---|---|---|---|---|---|---|
| IoU↑ | 0.998±0.001 | 0.964±0.075 | 0.959±0.064 | 0.910±0.066 | 0.816±0.127 | 0.521±0.145 | **0.861±0.169** |
| Dice/F1↑ | 0.999±0.001 | 0.980±0.050 | 0.978±0.038 | 0.952±0.039 | 0.893±0.082 | 0.673±0.133 | **0.912±0.118** |

#### 4.5.3 LATENT ENCODED FUSION

Table 7 validates our latent fusion for degradation semantic encoding ("what"). By fusing luminance and chrominance representations with global priors via adaptive modulation, our method achieves a separation ratio of **1.56** and KNN accuracy of **0.932**. Frequency-domain degradations (blur, noise) show perfect classifi-

Table 7: Latent Prior Encoding Quality: Quantitative evaluation of degradation-aware representation learning across different degradation types

| Metric | Blur | Haze | Lowlight | Noise | Rain | Snow | GLOBAL |
|---|---|---|---|---|---|---|---|
| Sep↑ | 6.15 | 1.62 | 2.04 | 1.56 | 1.15 | 1.48 | **1.56** |
| Acc↑ | 1.000 | 0.950 | 0.950 | 1.000 | 0.886 | 0.889 | **0.932** |

cation (Acc = 1.0), while pixel-level and spatially-varying degradations maintain strong separability (Acc > 0.88, Sep > 1.15). These results demonstrate that our latent fusion captures degradation-specific characteristics without explicit prompts, guiding the decoder in determining "what" content to reconstruct.

### 4.5.4  RESTORATION WITH 3WD

We evaluate DAIR with 3WD for leveraging reasoning cues against attention-based baselines: self-attention (SA), cross-attention (CA) (Vaswani et al., 2017), and window attention (WA) (Liang et al., 2021). As shown in Table 8, our method achieves **26.90 dB PSNR** and **0.8565 SSIM**, outperforming the best baseline (WA) by **+3.48 dB** and **+0.0621 SSIM**. Critically, our linear complexity $O(HW)$ enables **5.2×**

Table 8: Comparison and restoration with 3WD.

| Method | Complexity | Memory (MB)↓ | FPS↑ | PSNR↑/SSIM↑ |
|---|---|---|---|---|
| CA | $O(H^2W^2)$ | - | - | 17.71/0.5765 |
| SA | $O(H^2W^2)$ | - | - | 19.05/0.5444 |
| WA | $O(M^2 \cdot HW)$ | 4519.90 | 8.77 | 23.42/0.7944 |
| **3WD (Ours)** | $O(HW)$ | 2045.94 | 45.69 | 26.90/0.8565 |
| *Improvement* | **Linear** | **-54.7%** | **+5.2×** | **+3.48/+0.0621** |

**speedup** and **54.7% memory reduction** compared to WA's quadratic $O(M^2 \cdot HW)$ complexity. We achieve 45.69 FPS with 4593.76 MB at HD 720p, with similar efficiency maintained at 2K resolution (20.45 FPS, 4573.29 MB). This validates that our 3WD reasoning mechanism effectively captures spatial-degradation dependencies without quadratic computational overhead.

### 4.6  ABLATION STUDY

We conducted an ablation study combining four challenging tasks (denoising, desnowing, deraining, LLIE) to evaluate the key components of DAIR: Latent Priors Encoding (LPE) ("which"), Degradation Map (DM) ("where"), Latent Fusion (LF) ("what"), and Restoration Block (3WD). As shown in Table 9, the inclusion of the proposed component progressively enhances performance, with DAIR achieving the highest average **PSNR of 26.90** and **SSIM of 0.8565**. Notably, removing Latent Priors re-

Table 9: Ablation study on key components. Tick (✓) indicates the component is used, cross (✗) indicates not used.

| Method | LP | LF | DM | 3WD | PSNR↑/SSIM↑ |
|---|---|---|---|---|---|
| Base Model | ✗ | ✗ | ✗ | ✗ | 20.22/0.6936 |
| Base + LP | ✓ | ✗ | ✗ | ✗ | 20.49/0.7025 |
| Base + LP + LF | ✓ | ✓ | ✗ | ✗ | 21.08/0.7208 |
| Base + LP + LF + DM | ✓ | ✓ | ✓ | ✗ | 21.77/0.7439 |
| DAIR w/o Latent Prior | ✗ | ✗ | ✓ | ✓ | 22.67/0.7689 |
| **DAIR** | ✓ | ✓ | ✓ | ✓ | 26.90/0.8565 |
| *Improvement* | | | | | **+6.68/+0.1629** |

sults in significantly worse performance (**-4.23 PSNR / -0.0876 SSIM**), highlighting its critical role in capturing detailed degradation descriptions, subsequently guiding the model in effectively segregating different types of degradations. Please see Appendix B for module-wise ablation and details.

### 4.7  REAL-WORLD IMPLICATIONS

Tables 10 and 11 evaluate DAIR's restoration quality through downstream tasks and real-world unseen restoration. For OD (Table 10), YOLOv12 tested on degraded MS-COCO images shows DAIR achieves **34.9 AP$_{50:95}$**, **37.3 AP$_{50}$**, and **36.4 AP$_{75}$**, outperforming PromptIR by **+4.0**, **+4.0**, and **+4.5**

Table 10: OD on restored images

| Method | Average Precision | | |
|---|---|---|---|
| | AP$_{50:95}$ | AP$_{50}$ | AP$_{75}$ |
| PromptIR | 30.9 | 33.3 | 31.9 |
| DFPIR | 27.9 | 30.0 | 29.3 |
| ADAIR | 28.9 | 31.2 | 30.2 |
| **DAIR (Ours)** | 34.9 | 37.3 | 36.4 |
| *Improvement* | **+4.0** | **+4.0** | **+4.5** |

Table 11: Real-world unseen restoration

| Method | NIQE↓/MUSIQ↑/BRISQUE↓ |
|---|---|
| PromptIR | 9.65/43.81/40.03 |
| DFPIR | 8.01/45.02/25.52 |
| ADAIR | 6.85/46.10/28.45 |
| **DAIR (Ours)** | 5.25/50.19/25.97 |
| *Improvement* | **-1.60/+4.09/+0.45** |

points, demonstrating preserved semantic content. For perceptual quality (Table 11), no-reference metrics across real-world unseen datasets show DAIR achieves best scores: **5.25 NIQE** (-1.60), **50.19 MUSIQ** (+4.09), and **25.97 BRISQUE** (+0.45), confirming superior perceptual quality and strong generalization to unseen out-of-distribution scenarios. See Appendix D for details.

## 5  CONCLUSION

We propose DAIR, a unified framework for AIR that tackles the challenges of unknown and compound degradations by learning degradation characteristics directly from the degraded image itself, eliminating the need for manual text or visual prompts. Guided by a reasoning paradigm based on "which features to use, where to focus while restoring, and what to restore", DAIR enables spatially adaptive restoration through a lightweight decoder that effectively integrates all prior information. Extensive experiments demonstrate that DAIR surpasses SOTA methods across six common and five compound degradation scenarios, while robustly handling unseen cases, highlighting its scalability and generalizability. Further details on implementation, analysis, unseen cases, downstream vision tasks, and additional results are provided in the appendix.

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

# APPENDIX

## A NETWORK DETAILS

### A.1 MOTIVATION

A major challenge in all-in-one image restoration is encoding degradation-specific information in a compact yet generalizable form while recovering clean images $\mathbf{I}$ from degraded observations $\mathbf{I}_{deg} = \mathcal{D}_R(\mathbf{I}) + \epsilon$, where degradations $\mathcal{D}_R$ may arise from diverse sources. Current methods rely on explicit degradation cues, limiting their ability to handle unknown or compositional degradations.

From an information-theoretic perspective, effective restoration requires learning the conditional distribution $p(\mathbf{I}|\mathbf{I}_{deg})$ by disentangling degradation-specific information from content-preserving features. We model this through a variational framework where degradation characteristics are encoded in latent variable $\mathbf{z} \sim p(\mathbf{z}|\mathbf{I}_{deg})$. The VAE objective $\mathcal{L} = -\mathbb{E}_{q_\phi(\mathbf{z}|\mathbf{I}_{deg})}[\log p_\theta(\mathbf{I}|\mathbf{z}, \mathbf{I}_{deg})] + \beta \cdot \text{KL}(q_\phi(\mathbf{z}|\mathbf{I}_{deg})\|p(\mathbf{z}))$ learns meaningful degradation representations while ensuring generalization through prior regularization. Degradations exhibit domain-specific characteristics: let $\mathbf{I}_{deg} = [L, C]$ where $L$ represents luminance and $C$ chrominance components. Structural degradations primarily affect $L$ while color distortions manifest in $C$. Factorizing the posterior as

$q_\phi(\mathbf{z}|\mathbf{I}_{deg}) = q_{\phi_L}(\mathbf{z}_L|L) \cdot q_{\phi_C}(\mathbf{z}_C|C)$ enables specialized degradation modeling, reducing cross-domain interference and improving restoration through $p_\theta(\mathbf{I}|\mathbf{z}_L, \mathbf{z}_C, \mathbf{I}_{deg})$. We propose a hybrid VAE with separate LC encoders that automatically infers continuous degradation latent without explicit MP.

## A.2 VAE DESIGN

**Architecture.** To effectively integrate the VAE into our framework, we design a hybrid U-Net encoder–decoder with multi-head self-attention (Tian et al., 2025a) at the bottleneck. In particular, both the hybrid VAE and the reconstruction network incorporate the same spatial feature dimensions, enabling seamless interaction between latent representations and restoration features. The encoder progressively downsamples RGB images through four stages with channel dimensions $[40, 80, 160, 320]$ using 3×3 convolutions (stride=2 for downsampling, padding=1). Each stage contains a Residual Attention Block (He et al., 2016) combining spatial convolutions with channel attention (Hu et al., 2018) (reduction ratio 8):

$$\text{ResAttn}(\mathbf{f}) = \mathbf{f} + \text{Conv}_{3\times3}(\text{ReLU}(\text{Conv}_{3\times3}(\mathbf{f}))) \odot \sigma(\text{Conv}_{1\times1}(\text{GAP}(\mathbf{f}))) \tag{11}$$

The bottleneck applies 2 layers of Multi-Head Self-Attention (4 heads, $d_k = 80$) to the deepest features $\mathbf{x}_3 \in \mathbb{R}^{H/8 \times W/8 \times 320}$:

$$\mathbf{Q}, \mathbf{K}, \mathbf{V} = \mathbf{x}_3 \mathbf{W}^Q, \mathbf{x}_3 \mathbf{W}^K, \mathbf{x}_3 \mathbf{W}^V \tag{12}$$

$$\text{MHSA}(\mathbf{x}_3) = \text{softmax}\left(\frac{\mathbf{Q}\mathbf{K}^T}{\sqrt{80}}\right)\mathbf{V} \tag{13}$$

Latent parameters are computed as $\boldsymbol{\mu}, \log\boldsymbol{\sigma}^2 = \text{Conv}_{1\times1}(\text{MHSA}^2(\mathbf{x}_3))$ with latent dimension 320 and reparameterization $\mathbf{z} = \boldsymbol{\mu} + \boldsymbol{\sigma} \odot \boldsymbol{\epsilon}$. The decoder mirrors the encoder using transposed convolutions (stride=2) and identical ResAttn blocks, producing reconstructions $\hat{\mathbf{I}}_{deg} = \tanh(\text{Conv}(\text{Decoder}(\mathbf{z})))$.

**Objective Function.** We train the complete VAE architecture using a multi-component loss function:

$$\mathcal{L}_{\text{VAE}} = \mathcal{L}_{\text{recon}} + \beta\mathcal{L}_{\text{KL}} + \lambda_{\text{con}}\mathcal{L}_{\text{SupCon}} \tag{14}$$

where $\beta$ and $\lambda_{\text{con}}$ are balancing hyperparameters for different loss components to stabilize VAE training. We set $\beta = 0.3$ to provide strong KL regularization ensuring proper latent space structure, and $\lambda_{\text{con}} = 0.01$ to enable weak supervision for degradation separation at the deepest encoder level (pre-latent space).

The reconstruction loss employs L1 distance to ensure pixel-level fidelity:

$$\mathcal{L}_{\text{recon}} = \|\mathbf{I}_{deg} - \hat{\mathbf{I}}_{deg}\|_1 \tag{15}$$

where $\mathbf{I}_{deg}$ is the input degraded image and $\hat{\mathbf{I}}_{deg} = D_\theta(\mathbf{z})$ is the reconstructed output from the decoder.

The KL divergence (Kullback & Leibler, 1951) term regularizes the latent distribution:

$$\mathcal{L}_{\text{KL}} = \text{KL}(q_\phi(\mathbf{z}|\mathbf{I}_{deg})\|\mathcal{N}(\mathbf{0}, \mathbf{I})) \tag{16}$$

This loss term encourages the learned latent distribution $q_\phi(\mathbf{z}|\mathbf{I}_{deg})$ to be close to a standard normal distribution $\mathcal{N}(\mathbf{0}, \mathbf{I})$, ensuring that the latent space has good sampling properties and prevents overfitting.

To encourage discriminative latent representations, we incorporate weakly Supervised Contrastive Learning (Guo et al., 2023) on the spatially-pooled encoder:

$$\mathcal{L}_{\text{SupCon}} = -\sum_i \log \frac{\sum_{j \in P_i} \exp(\mathbf{z}_i \cdot \mathbf{z}_j / \tau)}{\sum_{k \neq i} \exp(\mathbf{z}_i \cdot \mathbf{z}_k / \tau)} \tag{17}$$

where $\mathbf{z}_i$ are the normalized features, $P_i$ represents samples with the same degradation label as sample $i$, and $\tau = 0.1$ is the temperature parameter. This loss pulls together samples from the same degradation category while pushing apart samples from different categories in the latent space, encouraging the encoder to learn discriminative representations for different types of image degradations.

**Capture degradation with VAE.** Fig. 8 illustrates the proposed VAE's progressive convergence and degradation-aware representation learning (Step = $0 \rightarrow 100k$). Incorporating SupCon loss as weak supervision improves latent separability of corruption types while preserving reconstruction fidelity, enabling the model to disentangle mixed degradations and generalize beyond the training distribution. Notably, it separates the unseen "Low + Rain" (Guo et al., 2024) from the related seen "Low + Haze + Rain," and characterizes out-of-domain cases such as underwater images, evidencing robust clustering and stable training dynamics.

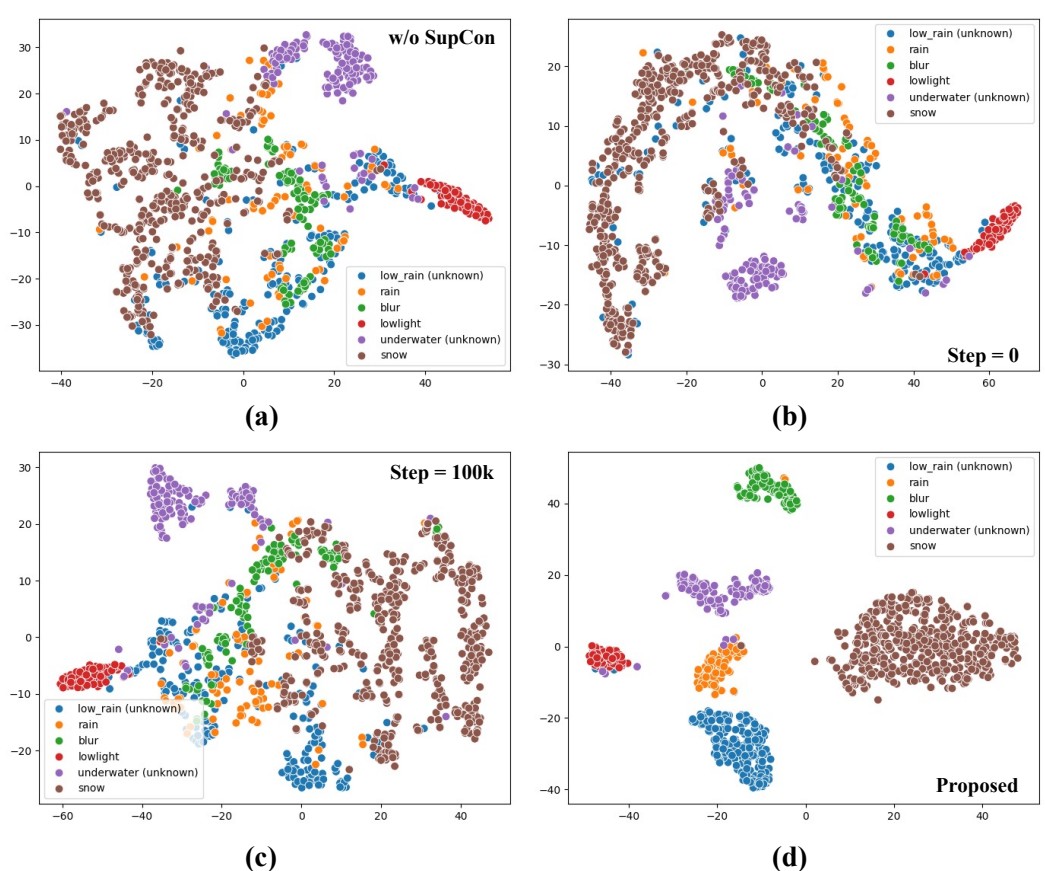

Figure 8: Learning degradation with VAE. (a) VAE without SupCon. (b) VAE at Step = 0. (c) VAE at Step = 100k. (d) VAE (fully trained).

## A.3 DAIR DETAILS

**Architecture.** We employ a dual-branch encoder with separate LC processing. RGB inputs are decomposed as $L = 0.299R + 0.587G + 0.114B$, $C = RGB - L$. Illumination and chrominance estimators utilize depthwise convolutions (kernel size=5, groups=4) to generate adaptive maps. Input embeddings are computed as $\mathbf{f}_{lum} = \text{LeakyReLU}(\text{Conv}_{3\times3}(\mathbf{I}_{deg} \cdot \text{IlluMap} + \mathbf{I}_{deg}))$ and $\mathbf{f}_{chrom} = \text{LeakyReLU}(\text{Conv}_{3\times3}(\mathbf{I}_{deg} \cdot \text{ChromMap} + \mathbf{I}_{deg}))$.

The encoder processes four levels with channel dimensions progressing as $40 \rightarrow 80 \rightarrow 160 \rightarrow 320$ for both LC branches. Downsampling uses $\text{Conv}_{3\times3}$ with stride=2 and padding=1, followed by LeakyReLU (slope=0.1). The decoder mirrors this structure using $\text{ConvTranspose}_{2\times2}$ (stride=2) for

upsampling $320 \rightarrow 160 \rightarrow 80 \rightarrow 40$. Final reconstruction applies $\text{Conv}_{3\times3}(\text{decoder\_out}) + \mathbf{I}_{deg}$ with tanh activation. All convolutions use padding=1 for embedding layers and one attention block per level.

**Objective Function.** The restoration network loss combines L1 reconstruction loss with SSIM-based perceptual loss:

$$\mathcal{L}_{\text{recon}} = \|\hat{\mathbf{I}} - \mathbf{I}\|_1 + \lambda_{\text{ssim}} \left( 1 - \text{SSIM} \left( \frac{\hat{\mathbf{I}} + 1}{2}, \frac{\mathbf{I} + 1}{2} \right) \right) \tag{18}$$

where images are normalized from $[-1, 1]$ to $[0, 1]$ range for SSIM computation, and $\lambda_{\text{ssim}} = 1.0$. The L1 loss ensures pixel-level fidelity while SSIM preserves perceptual quality and structural information. This combination effectively handles both fine-grained details and global image structure during restoration.

### A.4 TRAINING DETAILS

Algorithm 1 details the complete training procedure. Notably, the VAE is trained only once using known degradation types, and the same pre-trained weights are utilized across all experiments, including single-task restoration, unseen degradation scenarios, and ablation studies, demonstrating the framework's practical versatility in handling multi-degradation restoration tasks.

---

**Algorithm 1** DAIR training for AIR

---

**Require:** Degraded images $\{\mathbf{I}_{deg,i}\}$, clean images $\{\mathbf{I}_i\}$, labels $\{y_i\}$
**Ensure:** Trained VAE $\theta_{\text{VAE}}$, restoration network $\theta_{\text{REST}}$
1: Initialize HybridVAE($\theta_{\text{VAE}}$), RestNet($\theta_{\text{REST}}$)
2: Set $T_1 = 200\text{K}$, $T_2 = 500\text{K}$, $\beta_{\max} = 0.3$, $\lambda_{\text{con}} = 0.01$
3: **Phase 1: VAE Pretraining**
4: **for** $t = 1$ to $T_1$ **do**
5: $\quad \hat{\mathbf{I}}_{deg}, \boldsymbol{\mu}, \log \boldsymbol{\sigma}^2, \mathbf{x}_0, \mathbf{x}_1, \mathbf{x}_2, \mathbf{x}_3 \leftarrow \text{VAE}(\mathbf{I}_{deg})$
6: $\quad \hat{\mathbf{z}}_3 \leftarrow \text{L2Norm}(\text{GAP}(\mathbf{x}_3))$ {Contrastive features}
7: $\quad \mathcal{L}_{\text{recon}} = \|\hat{\mathbf{I}}_{deg} - \mathbf{I}_{deg}\|_1$
8: $\quad \mathcal{L}_{\text{KL}} = -\frac{1}{2} \sum (1 + \log \boldsymbol{\sigma}^2 - \boldsymbol{\mu}^2 - \boldsymbol{\sigma}^2)$
9: $\quad \mathcal{L}_{\text{SupCon}} = \text{SupConLoss}(\hat{\mathbf{z}}_3, y)$
10: $\quad \beta(t) = \min(\beta_{\max}, \beta_{\max} \cdot t/T_1)$ {KL annealing}
11: $\quad \mathcal{L}_{\text{VAE}} = \mathcal{L}_{\text{recon}} + \beta(t)\mathcal{L}_{\text{KL}} + \lambda_{\text{con}}\mathcal{L}_{\text{SupCon}}$
12: $\quad$ Update $\theta_{\text{VAE}}$ via $\nabla_{\theta_{\text{VAE}}} \mathcal{L}_{\text{VAE}}$
13: **end for**
14: **Phase 2: Restoration Network Training**
15: Freeze($\theta_{\text{VAE}}$) {Fix VAE parameters}
16: **for** $t = T_1 + 1$ to $T_1 + T_2$ **do**
17: $\quad \boldsymbol{\mu}, \mathbf{x}_0, \mathbf{x}_1, \mathbf{x}_2, \mathbf{x}_3 \leftarrow \text{VAE}(\mathbf{I}_{deg})$ {No gradients}
18: $\quad \hat{\mathbf{I}} \leftarrow \text{RestNet}(\mathbf{I}_{deg}, \mathbf{x}_0, \mathbf{x}_1, \mathbf{x}_2, \mathbf{x}_3, \boldsymbol{\mu})$
19: $\quad \mathcal{L}_{\text{recon}} = \|\hat{\mathbf{I}} - \mathbf{I}\|_1 + (1 - \text{SSIM}(\hat{\mathbf{I}}, \mathbf{I}))$
20: $\quad$ Update $\theta_{\text{REST}}$ via $\nabla_{\theta_{\text{REST}}} \mathcal{L}_{\text{recon}}$
21: **end for**
22: **return** $\theta_{\text{VAE}}, \theta_{\text{REST}}$

---

## B ANALYSIS AND ABLATION OF DAIR

We conduct an ablation and module-wise analysis on four common tasks: denoising, desnowing, deraining, and LLIE. This section provides a detail the performance and contributions of our proposed module across these tasks.

## B.1 LATENT PRIOR ENCODING ("WHICH")

We analyze the "which" component and demonstrate that LC separation makes feature selection both explicit and effective. By routing structure-dominant cues through luminance and color-specific cues through chrominance, the encoder preserves complementary statistics that can be modulated by latent priors per branch. This design validates our reasoning by enabling degradation-aware, stage-wise selection of "which" features to leverage, resulting in substantial gains over luminance-only, chrominance-only, and RGB based encoders, as shown in Table 12.

Table 12: Ablation study of which module.

| Method | PSNR↑/SSIM↑/LPIPS↓ |
|---|---|
| Luma only | 23.58/0.7996/0.1946 |
| Chroma only | 23.41/0.7927/0.2014 |
| RGB | 23.44/0.8002/0.1892 |
| **Proposed (Both)** | **26.90/0.8565/0.1316** |
| *Improvement* | **+3.32/+0.0569/-0.0630** |

## B.2 DEGRADATION MAP ("WHERE")

We evaluate the impact of the degradation map ("where") separately. Table 13 demonstrates that the proposed degradation map with the FFT module significantly improves restoration performance. The model without the degradation map achieved a PSNR of 22.80, SSIM of 0.7769, and LPIPS of 0.2116, whereas our approach with the degradation map improved these metrics to a PSNR of 26.90, SSIM of 0.8565, and LPIPS of 0.1316, resulting in enhancements of +4.10 in PSNR, +0.0796 in SSIM, and a reduction of -0.0800 in LPIPS. It also ensures that the degradation map captures frequency-domain degradation characteristics, leveraging frequency cues.

Table 13: Ablation study of degradation map ("where") module.

| Method | PSNR↑/SSIM↑/LPIPS↓ |
|---|---|
| w/o FFT | 22.80/0.7769/0.2116 |
| **Proposed** | **26.90/0.8565/0.1316** |
| *Improvement* | **+4.10/+0.0796/-0.0800** |

## B.3 LATENT FUSION ("WHAT")

We systematically evaluate the impact of different fusion strategies on restoration performance. Table 14 illustrates the results of latent fusion across various settings. Our full $\mu$-fusion (Factor=1) achieves a PSNR of 26.90 dB, outperforming the second-best method by +3.92 dB. Notably, removing degradation-aware components (beta: 22.97 dB, MU-fusion: 22.74 dB) or employing fixed fusion factors (Factor=0: 17.88 dB) results in significant performance drops. This underscores that the learned $\mu$-fusion weights effectively capture degradation characteristics and facilitate semantically meaningful decisions.

Table 14: Ablation study of fusion strategies. Best in **bold red**, second underlined.

| Method | Beta | Alpha | Factor | PSNR↑/SSIM↑/LPIPS↓ |
|---|---|---|---|---|
| w/o MU | ✗ | ✗ | ✗ | 22.74/0.7791/0.2135 |
| w/o beta | ✗ | ✓ | 1 | 22.97/0.7824/0.2147 |
| alpha + beta | ✓ | ✓ | 0 | 17.88/0.7272/0.2488 |
| Fusion 0.1 | ✓ | ✓ | 0.1 | 16.28/0.7350/0.2345 |
| Fusion 0.5 | ✓ | ✓ | 0.5 | 22.98/0.7837/0.2143 |
| **Proposed** | ✓ | ✓ | 1 | **26.90/0.8565/0.1316** |
| *Improvement* | | | | **+3.92/+0.0728/-0.0827** |

### B.4 Detail Ablation

We provide a module-wise detailed ablation study of the proposed DAIR in Table 15. This table illustrates the contributions of key components to the overall restoration performance. Each row represents a different configuration, with tick marks (✓) indicating the inclusion of specific components and crosses (✗) indicating their exclusion. The best results are highlighted in **bold red**. Notably, the full DAIR model demonstrates superior performance across all metrics, validating the effectiveness of the integrated components.

Table 15: Ablation study on key components.

| Method | LP | LF | DM | 3WD | Denoise PSNR↑/SSIM↑ | Desnowing PSNR↑/SSIM↑ | Derain PSNR↑/SSIM↑ | Lowlight PSNR↑/SSIM↑ |
|---|---|---|---|---|---|---|---|---|
| Base Model | ✗ | ✗ | ✗ | ✗ | 23.42/0.724 | 20.76/0.770 | 22.70/0.770 | 13.99/0.511 |
| Base + LP | ✓ | ✗ | ✗ | ✗ | 24.50/0.742 | 20.75/0.778 | 22.46/0.766 | 14.23/0.524 |
| Base + LP + LF | ✓ | ✓ | ✗ | ✗ | 25.83/0.780 | 21.21/0.801 | 22.23/0.755 | 15.04/0.547 |
| Base + LP + LF + DM | ✓ | ✓ | ✓ | ✗ | 26.80/0.816 | 21.97/0.844 | 22.79/0.761 | 15.52/0.554 |
| DAIR w/o Latent Prior | ✗ | ✗ | ✓ | ✓ | 26.07/0.886 | 26.83/0.831 | 22.76/0.774 | 15.00/0.585 |
| **DAIR** | ✓ | ✓ | ✓ | ✓ | **29.11/0.897** | **31.59/0.958** | **30.21/0.907** | **16.68/0.664** |

## C Additional Comparison with SOTA Methods

### C.1 Comparison on Three-task Settings

Our primary motivation is to automate the prompting process and reframe AIR for practical, real-world applications. In real-world scenarios, degraded scenes are often highly complex and frequently involve overlapping degradations. To demonstrate the effectiveness of our method in addressing both homogeneous and heterogeneous degradations, we evaluate its practicality in a challenging six-task heterogeneous setting comprising complex datasets (LSD for real-world lowlight (Sharif et al., 2025), random denoising instead of fixed denoising, heavy rain streak, etc.). We refer to these tasks as a common task in the manuscript. Furthermore, we incorporate a compound 5D task, where degradations are similar and overlapping (e.g., Haze + Snow, Haze + Lowlight + Snow). Thus, we can illustrate the limitations of existing work and clearly articulate our motivation. However, in recent times, many methods, including PromptIR, ADAIR, DFPIR, Perceive-IR, etc., have leveraged three task settings to illustrate restoration performance. Therefore, to establish a clear positioning of our proposed method relative to existing models, we benchmarked it under similar settings. Table 16 compares our DAIR with existing methods. Our proposed method outperforms all evaluated metrics, achieving notable improvements of +2.50 dB in dehazing, +0.39 dB in denoising at $\sigma = 15$, and +3.80 dB in denoising at $\sigma = 50$ compared to the best baseline. These results underscore the effectiveness of DAIR in enhancing restoration quality.

Table 16: Comparison of image restoration methods across different tasks and noise levels. Best results in **bold red**, second best underlined, and increment over best baseline highlighted in **blue**.

| Method | Dehazing SOTS PSNR↑/SSIM↑ | Deraining Rain100L PSNR↑/SSIM↑ | Denoising (CBSD68) $\sigma = 15$ PSNR↑/SSIM↑ | $\sigma = 25$ PSNR↑/SSIM↑ | $\sigma = 50$ PSNR↑/SSIM↑ | Average PSNR↑/SSIM↑ |
|---|---|---|---|---|---|---|
| PromptIR | 30.58/0.974 | 36.37/0.972 | 33.98/0.933 | 31.31/0.888 | 28.06/0.799 | 32.06/0.913 |
| Restormer | 30.43/0.975 | 36.55/0.975 | 33.84/0.931 | 31.18/0.885 | 27.90/0.790 | 31.98/0.911 |
| DFPIR | 31.87/0.980 | 38.65/0.982 | 34.14/0.935 | 31.47/0.893 | 28.25/0.806 | 32.88/0.919 |
| ADAIR | 31.06/0.980 | 38.64/0.983 | 34.12/0.935 | 31.45/0.892 | 28.19/0.802 | 32.69/0.918 |
| **DAIR(Ours)** | **34.37/0.987** | **38.51/0.985** | **34.51/0.947** | **31.99/0.913** | **31.99/0.913** | **34.27/0.949** |
| *Improvement* | +2.50/+0.007 | -0.13/+0.002 | +0.39/+0.012 | +0.54/+0.021 | +3.80/+0.111 | +1.58/+0.031 |

### C.2 Common Task Details

#### C.2.1 Low-light

We utilize the LSD dataset (Sharif et al., 2025), collected in uncontrolled low-light settings, offering diverse indoor and outdoor scenes under varying conditions. Table 17 compares image restoration

methods across extreme lowlight (under 50 Lux) and lowlight scenarios (50-200 lux), divided into indoor and outdoor subsets, using PSNR and SSIM metrics. Our method, DAIR, consistently outperforms others, with the best results highlighted in bold red and the second-best underlined. DAIR achieves significant gains, improving up to +2.64 PSNR and +0.0737 SSIM in extreme lowlight, showcasing its robustness and superior generalization in challenging conditions.

Table 17: Performance comparison across lighting conditions. Best results in **bold red**, second best underlined.

| Method | Extreme Lowlight | | Lowlight | |
|---|---|---|---|---|
| | **Indoor**
PSNR↑/SSIM↑ | **Outdoor**
PSNR↑/SSIM↑ | **Indoor**
PSNR↑/SSIM↑ | **Outdoor**
PSNR↑/SSIM↑ |
| Uformer | 11.98/0.5278 | 13.24/0.4628 | 10.39/0.5535 | 12.79/0.4145 |
| Restormer | 14.99/0.6343 | 16.35/0.5828 | 16.07/0.7121 | 13.90/0.5429 |
| AIRNet | 8.51/0.3678 | 9.80/0.3248 | 8.13/0.4585 | 9.05/0.3107 |
| PromptIR | 14.74/0.5902 | 16.10/0.5038 | 14.59/0.6404 | 13.48/0.4154 |
| DiffUIR | 9.77/0.5185 | 11.86/0.3713 | 9.77/0.5185 | 11.29/0.3193 |
| ADAIR | 12.92/0.5456 | 14.46/0.4783 | 10.93/0.5576 | 12.72/0.4079 |
| DFPIR | 13.32/0.5643 | 15.68/0.4815 | 15.63/0.6599 | 14.74/0.3871 |
| **DAIR (Ours)** | **17.63/0.7080** | **17.94/0.6486** | **17.13/0.7180** | **14.77/0.5898** |
| *Improvement* | **+2.64/+0.0737** | **+1.59/+0.0658** | **+1.06/+0.0059** | **+0.87/+0.0469** |

### C.2.2 DENOISING

Our experiment assesses the generalization of image denoising models under variable noise conditions, addressing limitations of prior works that train on fixed noise levels, leading to overfitting. We retrained baseline methods and our DAIR model using random Gaussian noise on DIV2K (Agustsson & Timofte, 2017) and evaluated on unseen datasets, BSD100 (Martin et al., 2001) and Urban100 (Kim et al., 2016). Table 18 illustrate performance comparison across these datasets and noise levels ($\sigma = 15, 25, 50$). Notably, baseline models showed inconsistent performance; Restormer struggled at higher noise, DiffUIR (Zheng et al., 2024) excelled on BSD100 at $\sigma$=50 but failed elsewhere, on six-task common AIR settings. Notably, MP-based methods, such as DFPIR (Tian et al., 2025a), underperformed in the absence of precise noise-level prompts. This inconsistent performance, particularly among MP-based methods, underscores their inability to effectively segregate degradations, even on homogeneous tasks. In contrast, DAIR consistently achieved higher scores, with improvements of up to +2.2 dB PSNR at high noise levels, highlighting its superior robustness and ability to effectively understand degradations, thereby validating the effectiveness of its latent prior encoding.

Table 18: Performance comparison across datasets and noise levels. Best results in **bold red**, second best underlined.

| Method | DIV2K | | | BSD100 | | | Urban100 | | |
|---|---|---|---|---|---|---|---|---|---|
| | $\sigma$=15
PSNR↑/SSIM↑ | $\sigma$=25
PSNR↑/SSIM↑ | $\sigma$=50
PSNR↑/SSIM↑ | $\sigma$=15
PSNR↑/SSIM↑ | $\sigma$=25
PSNR↑/SSIM↑ | $\sigma$=50
PSNR↑/SSIM↑ | $\sigma$=15
PSNR↑/SSIM↑ | $\sigma$=25
PSNR↑/SSIM↑ | $\sigma$=50
PSNR↑/SSIM↑ |
| Uformer | 31.71/0.9265 | 29.16/0.8753 | 25.46/0.7351 | 33.28/0.9219 | 30.61/0.8635 | 33.28/0.9219 | 31.52/0.9387 | 29.13/0.8944 | 25.15/0.7664 |
| Restormer | 27.38/0.8567 | 23.44/0.6825 | 16.62/0.3646 | 28.32/0.8385 | 23.73/0.6373 | 31.59/0.8937 | 28.11/0.8873 | 23.67/0.7304 | 16.52/0.4315 |
| AIRNet | 32.91/0.9377 | 30.52/0.9095 | 27.07/0.8307 | 34.20/0.9333 | 31.83/0.8986 | 27.03/0.7006 | 33.19/0.9506 | 31.00/0.9324 | 27.46/0.8665 |
| PromptIR | 28.84/0.8464 | 25.90/0.7306 | 21.45/0.5164 | 30.04/0.8295 | 27.03/0.7007 | 30.61/0.8636 | 29.23/0.8755 | 26.05/0.7734 | 21.23/0.5685 |
| DiffUIR | 19.92/0.8065 | 15.95/0.6774 | 14.34/0.5435 | 21.78/0.7909 | 17.41/0.6917 | 31.99/0.9122 | 18.28/0.7967 | 14.86/0.6701 | 13.67/0.5496 |
| ADAIR | 32.51/0.9412 | 30.10/0.9075 | 26.80/0.8174 | 34.06/0.9354 | 31.59/0.8937 | 14.07/0.4531 | 32.74/0.9514 | 30.33/0.9233 | 26.72/0.8420 |
| DFPIR | 22.13/0.8095 | 21.57/0.7585 | 20.43/0.6234 | 21.77/0.8164 | 21.47/0.7535 | 21.48/0.7533 | 22.36/0.8182 | 21.80/0.7759 | 21.48/0.7533 |
| **DAIR (Ours)** | **33.08/0.9541** | **30.62/0.9252** | **27.25/0.8555** | **34.48/0.9468** | **31.99/0.9122** | **31.83/0.8988** | **33.23/0.9623** | **30.82/0.9395** | **27.30/0.8796** |
| *Improvement* | **+0.17/+0.0164** | **+0.10/+0.0157** | **+0.18/+0.0248** | **+0.28/+0.0135** | **+0.16/+0.0136** | **-0.16/-0.0134** | **+0.04/+0.0117** | **-0.18/+0.0071** | **-0.16/+0.0131** |

## C.3 MORE VISUAL RESULTS

Fig. 9 presents visual results for denoising, deraining, and dehazing tasks, comparing our method with ADAIR (Cui et al., 2024), DFPIR (Tian et al., 2025a), degraded inputs, and reference images. These examples demonstrate that our approach consistently achieves clearer and more accurate restorations, closely matching the reference images and outperforming baselines across all degradation types.

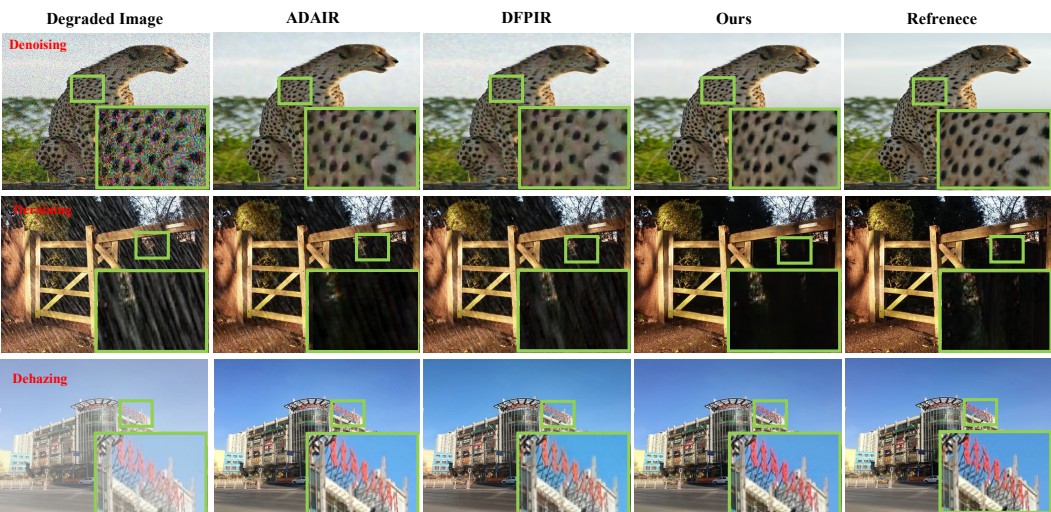

Figure 9: Visual comparisons of common image restoration, including denoising, deraining, and dehazing. Our method consistently delivers clearer and more accurate outputs, closely matching the reference images and outperforming ADAIR and DFPIR.

Fig. 10 presents more visual results for compound degradations (e.g., low-light + haze + rain, haze + rain), comparing our method with ADAIR (Cui et al., 2024), DFPIR (Tian et al., 2025a), degraded inputs, and reference images. Our approach consistently delivers clearer and more accurate restorations, outperforming baselines in challenging compound degradation settings.

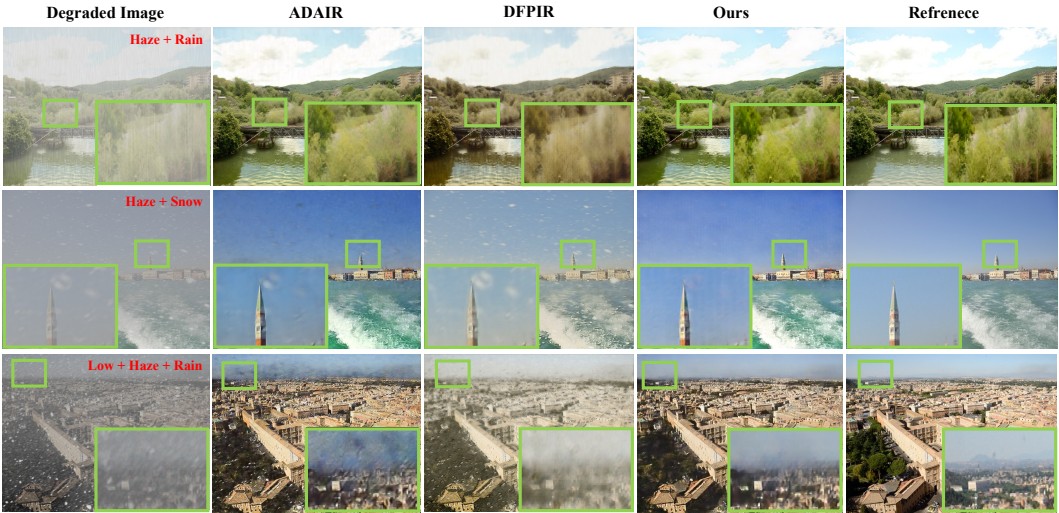

Figure 10: Visual comparisons of compound image restoration results for challenging scenarios, including haze + rain, haze + snow, and low-light + haze + rain. Our method consistently delivers clearer and more accurate outputs, closely matching the reference images and outperforming ADAIR and DFPIR.

## C.4 INFERENCE BENCHMARKING

Table 19 presents the inference speed of image restoration models, evaluated on lower-mid hardware (RTX 3060) using full-precision weights (FP32). Our DAIR model consistently achieves the fastest processing times, highest FPS, and superior throughput across various image sizes, outperforming all baseline AIR models. Notably, DAIR matches or surpasses the single-task model like Uformer

Table 19: Inference speed comparison across different image restoration models. Best results in **bold red**, second best underlined.

| Model | 128×128 | | | 256×256 | | | 512×512 | | |
|-------|---------|---|---|---------|---|---|---------|---|---|
| | Time (ms)↓ | FPS↑ | Throughput (MP/s)↑ | Time (ms)↓ | FPS↑ | Throughput (MP/s)↑ | Time (ms)↓ | FPS↑ | Throughput (MP/s)↑ |
| Uformer | 31.07 | 32.19 | 0.53 | 58.08 | 17.22 | 1.13 | 203.83 | 4.91 | 1.29 |
| PromptIR | 65.62 | 15.24 | 0.25 | 180.31 | 5.55 | 0.36 | 707.17 | 1.41 | 0.37 |
| ADAIR | 71.01 | 14.08 | 0.23 | 191.50 | 5.22 | 0.34 | 732.94 | 1.36 | 0.36 |
| DFPIR (with CLIP) | 70.08 | 14.27 | 0.23 | 184.33 | 5.43 | 0.36 | 714.20 | 1.40 | 0.37 |
| DFPIR (dummy text) | 67.88 | 14.73 | 0.24 | 184.12 | 5.43 | 0.36 | 714.38 | 1.40 | 0.37 |
| **DAIR (Ours)** | **20.55** | **48.67** | **0.80** | **55.20** | **18.12** | **1.19** | 220.92 | 4.53 | 1.19 |

(**43.86G**) (Wang et al., 2022), which is significantly lighter compared to the existing AIR methods. These results highlight DAIR's efficiency, demonstrating its ability to deliver SOTA restoration quality while maintaining faster speed and practicality for real-world applications, even on modest hardware setups.

# D  MORE RESULTS FROM REAL-WORLD

## D.1  DOWNSTREAM VISION TASKS (OD)

A primary objective of AIR is enhancing downstream vision task performance (Cui et al., 2024; Jiang et al., 2025), yet existing AIR methods lack benchmarking on such tasks due to insufficient annotated datasets. To address this critical gap, we created a comprehensive evaluation framework by annotating 3,000 test images across six common degradations (lowlight (Sharif et al., 2025), deraining (Fu et al., 2017), dehazing (Li et al., 2018), desnowing (Li et al., 2020), denoising (Agustsson & Timofte, 2017), and deblurring (Nah et al., 2017)). Our annotation process leveraged the paired nature of these datasets by first annotating ground truth reference images and transferring these annotations to their corresponding corrupted counterparts, ensuring consistency and reliability. We then enhanced these degraded images using several baseline AIR restoration methods and systematically evaluated their performance using the YOLOv12-Large (Tian et al., 2025b) OD model. This novel benchmarking approach provides the first quantitative assessment of how different restoration techniques impact OD performance, offering valuable insights for developing AIR methods optimized for real-world computer vision applications.

Table 20: Performance comparison of object detection models. Best results in **bold red**, second best underlined.

| Method | Average Precision (AP) | | | AP by Object Size | | |
|--------|------------------------|---|---|-------------------|---|---|
| | IoU=0.50:0.95 | IoU=0.50 | IoU=0.75 | Small | Medium | Large |
| Uformer | 28.0 | 30.2 | 29.4 | 36.5 | 26.9 | 27.3 |
| Restormer | 31.7 | 34.1 | 33.2 | 37.5 | 28.7 | 32.4 |
| AIRNet | 30.2 | 32.7 | 31.8 | 35.8 | 26.4 | 30.3 |
| PromptIR | 30.9 | 33.3 | 31.9 | 41.8 | 28.6 | 30.6 |
| ADAIR | 28.9 | 31.2 | 30.2 | 38.4 | 28.8 | 28.1 |
| DFPIR | 27.9 | 30.0 | 29.3 | 26.6 | 25.5 | 29.8 |
| **DAIR (Our)** | **34.9** | **37.3** | **36.4** | **44.9** | **30.8** | **35.1** |
| *Improvement* | **+3.2** | **+3.2** | **+3.2** | **+3.1** | **+2.0** | **+2.7** |

Table 20 presents a comprehensive performance comparison of various image restoration methods for OD tasks. Our proposed DAIR method demonstrates superior performance across all evaluation metrics, achieving the highest Average Precision (AP) scores of **34.9%** at IoU=0.50:0.95, **37.3%** at IoU=0.50, and **36.4%** at IoU=0.75, surpassing the second-best method (Restformer (Zamir et al., 2022)) by a significant margin of **3.2 percentage points**. DAIR also excels in detecting objects of varying sizes, achieving notably high AP scores on small (**44.9%**, outperforming PromptIR's **41.8%**), medium (**30.8%**), and large (**35.1%**) objects. The consistent improvements across all metrics highlight DAIR's effectiveness in enhancing image quality for downstream OD, with an **average improvement of 3.2 percentage points** over SOTA methods. These results demonstrate DAIR's robust capability to address diverse degradation scenarios while preserving essential visual information for accurate OD.

## D.2 Unseen Task Generalization

A key motivation behind the proposed latent-prior DAIR framework is its ability to generalize to diverse, previously unseen degradations. To validate this capability, we extensively evaluate DAIR under cross-domain and out-of-distribution scenarios, including underwater image enhancement (Peng & Cosman, 2019), medical image perceptual enhancement (Uhlen et al., 2010), medical image denoising (Rezvantalab et al., 2018), real-world denoising (Abdelhamed et al., 2018), and unseen real-world low-light enhancement (Sharif et al., 2025).

### D.2.1 Comparison with existing Methods.

We also compared existing methods against DAIR on unseen real-world degradation. We benchmarked all models trained on common degradation without any task-specific fine-tuning. The quantitative scores are summarized using no-reference metrics: NIQE Mittal et al. (2013), MUSIQ Ke et al. (2021), and LPIPS Zhang et al. (2018b). Table 21 presents a performance comparison on no-reference quality metrics. Our proposed method, DAIR, achieves the best results in most categories, demonstrating significant improvements, including a reduction of -0.78 in NIQE for unseen Real-Lowlight and an increase of +5.08 in MUSIQ. Furthermore, DAIR excels in Real-Noise and Real Perceptual Enhancement tasks, highlighting its effectiveness in enhancing image quality across a range of applications.

Table 21: Performance comparison across different image restoration tasks using no-reference quality metrics. Best results in **bold red**, second best underlined, and increment over best performing method highlighted in **blue**.

| Method | Real-Lowlight NIQE↓/MUSIQ↑/BRISQUE↓ | Real-Noise NIQE↓/MUSIQ↑/BRISQUE↓ | Real Underwater NIQE↓/MUSIQ↑/BRISQUE↓ | Real Perceptual Enhancement NIQE↓/MUSIQ↑/BRISQUE↓ | Medical Denoising NIQE↓/MUSIQ↑/BRISQUE↓ | Average NIQE↓/MUSIQ↑/BRISQUE↓ |
|---|---|---|---|---|---|---|
| Input | 5.33/47.30/27.17 | 4.44/62.56/5.55 | 3.83/52.35/**8.80** | 13.94/37.06/56.26 | 17.06/21.81/65.10 | 8.92/44.42/32.58 |
| PromptIR | 5.31/42.58/39.07 | 5.77/61.11/27.38 | 3.98/54.00/17.45 | 7.59/42.62/38.14 | 25.60/18.72/78.09 | 9.65/43.81/40.03 |
| DFPIR | 5.50/45.81/21.83 | 4.88/59.72/16.59 | 4.06/52.16/8.97 | 9.55/43.13/**15.52** | 16.04/24.27/64.71 | 8.01/45.02/25.52 |
| ADAIR | 5.53/46.71/35.77 | 4.38/63.69/16.30 | 3.63/**55.50**/13.29 | 11.08/40.27/33.54 | 9.65/24.34/43.36 | 6.85/46.10/28.45 |
| **DAIR (Ours)** | **4.75/51.79/21.97** | **4.24/69.37/16.94** | **3.61**/55.32/15.50 | **5.42/45.94**/34.91 | **8.22/28.53/24.55** | **5.25/50.19/25.97** |
| *Improvement* | *-0.78/+5.08/+0.14* | *-0.14/+5.68/+0.35* | *-0.02/-0.18/+6.70* | *-2.17/+2.81/+19.39* | *-1.43/+4.19/-18.81* | *-1.60/+4.09/+0.45* |

### D.2.2 Visual results

Figure 11 illustrate the performance of the proposed method on real-world noisy images (Abdelhamed et al., 2018). In well-exposed scenes, DAIR performs only denoising, preserving natural brightness and contrast. However, when inputs exhibit low-light characteristics (as commonly observed in SSID-like datasets (Abdelhamed et al., 2018)) along with noise, the latent descriptor guides DAIR to jointly suppress noise and enhance brightness. Thus, DAIR adaptively performs denoising and enhancement without prompts or handcrafted rules.

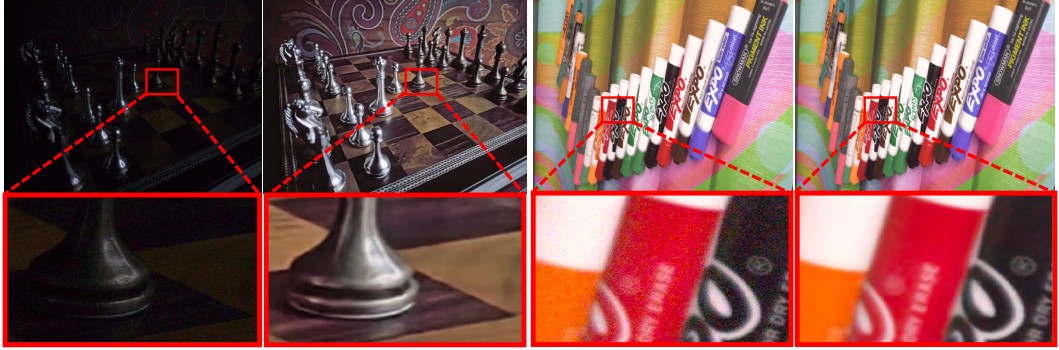

Figure 11: Real-world unseen degradation restoration with DAIR. Our method can restore unseen out-of-distribution datasets without needing any fine tuning.

Similarly, our method can significantly improve the visual quality of unseen diverse real-world degradation, as shown in Fig. 12

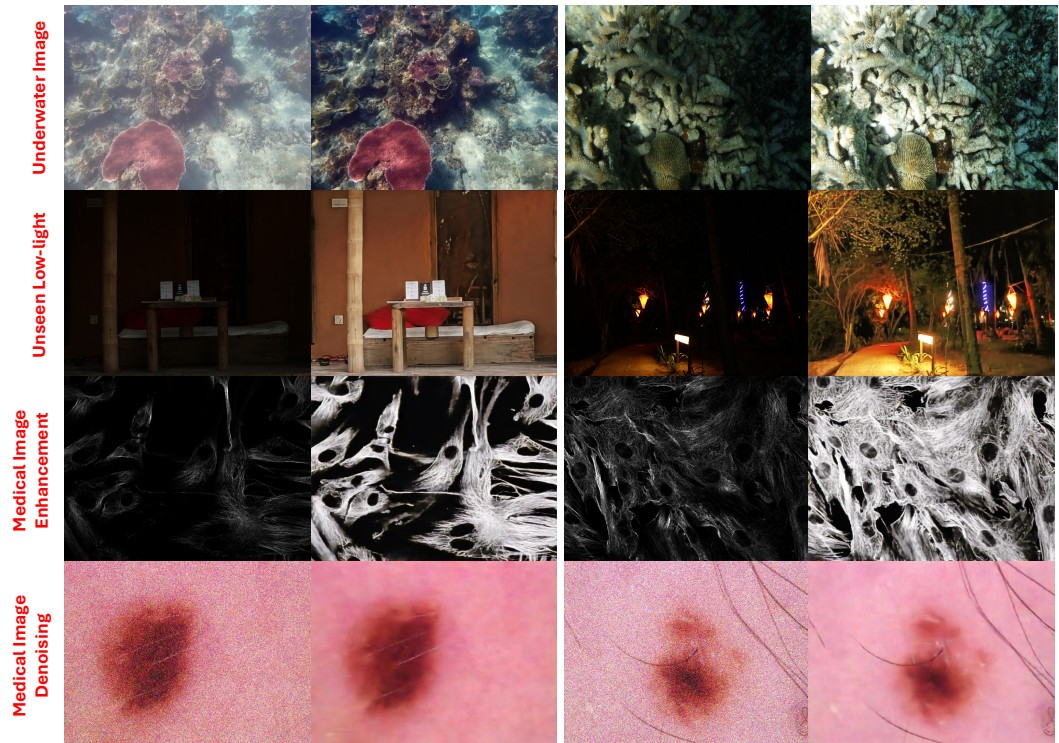

Figure 12: Real-world unseen degradation restoration with DAIR. Our method can restore unseen out-of-distribution datasets without needing any fine tuning.

Table 22: Quantitative comparison on synthesized and real datasets. Best results in **bold red**.

| Model | Training | Synthesized | | | Real | | |
|---|---|---|---|---|---|---|---|
| | | PSNR↑ | SSIM↑ | LPIPS↓ | NIQE↓ | MUSIQ↑ | BRISQUE↓ |
| Low (input) | - | 8.74 | 0.4255 | 0.4943 | 7.04 | 32.18 | 36.22 |
| ADAIR | Zero-shot | 11.77 | 0.5528 | 0.4057 | 6.80 | 33.62 | 32.57 |
| **DAIR** | Zero-shot | **13.35** | **0.6221** | **0.3393** | **5.14** | **46.39** | **28.58** |
| ADAIR | Fine-tune | 22.83 | 0.7811 | 0.2406 | 6.80 | 33.62 | 32.57 |
| **DAIR** | Fine-tune | **24.10** | **0.8532** | **0.1503** | **4.70** | **49.58** | **27.74** |

### D.3 REAL-WORLD COMPOUND DEGRADATION

We evaluated our method on real-world compound degradation (i.e., LOL-Blur (Zhou et al., 2022). Table 22 presents a quantitative comparison of image restoration models on synthesized and real datasets. The results demonstrate that our proposed method, DAIR, outperforms existing models in all evaluated metrics. Notably, DAIR achieves the highest PSNR and SSIM values, as well as the lowest LPIPS and NIQE scores, indicating superior restoration quality compared to ADAIR. DAIR achives a PSNR of **24.10** and an SSIM of **0.8532** few-shot fine tunning. Fig. 13 further demonstrates the practicality of our method through a visual comparison.

## E   LIMITATION AND FUTURE SCOPE

Despite the significant improvements of our DAIR method over existing approaches for both common and compound degradations, we identified some limitations. In real-world compound scenes, such as those in the LOL-Blur dataset(Zhou et al., 2022), our method can produce artifacts. Figure 14 shows failure cases on unseen tasks. Specifically, in extreme low-light conditions (under 5 lux), artifacts were observed in homogeneous spatial regions, such as cloud-free skies. Addition-

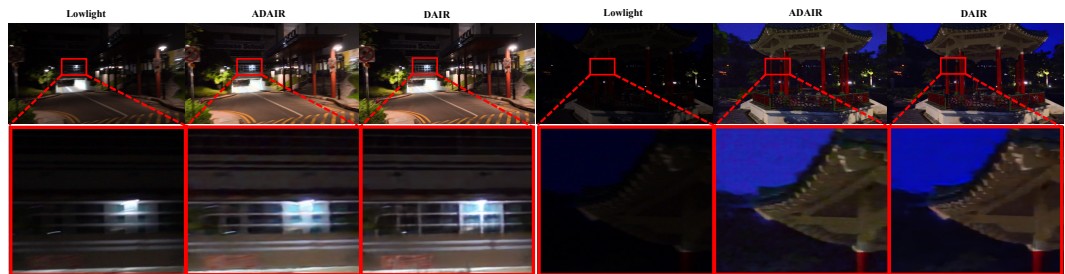

Figure 13: Real-world compound degradation restoration with DAIR.

ally, in underwater image enhancement, the method occasionally produced over-enhanced results in tricky scenes. These observations highlight areas that require further investigation, which we plan to address in future studies.

Moreover, the proposed method was tested on lower-mid desktop environments. In contrast, AIR demonstrates significant potential for deployment on edge hardware. As part of future work, we plan to optimize DAIR for edge hardware by employing techniques such as low-bit quantization and mixed-precision inference. Furthermore, integrating vision tasks with AIR for practical applications presents another exciting direction for future research on DAIR.

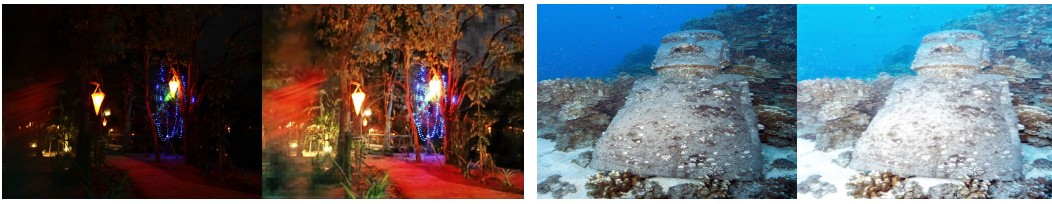

Figure 14: Example of failure cases. Left: visual artifacts on the spatial region in extreme low-light conditions, right: over-enhancement of the underwater image.

