# OpenReview forum: "Degradation-Aware All-in-One Image Restoration via Latent Prior Encoding"
_ICLR.cc/2026/Conference — Submitted to ICLR 2026_

### Official Review · Reviewer_SuKu · 2025-10-26

**Soundness:** 2
**Presentation:** 4
**Contribution:** 3
**Rating:** 4
**Confidence:** 4

**Summary:**

The paper proposes DAIR, a degradation aware all-in-one restoration approach which aims to tackle unknown degradations. Existing methods require human manual inputs, prompts or hand-crafted priors to provide degradation information to the model, which presents challenges when dealing with unseen degradations. DAIR is built to automatically tackle unseen and compound degradations with a 'which, where, what' approach. The authors propose latent prior encoding to identify 'which' features from the encoder are required for restoration. A degradation map is then constructed which dictates 'where' the model should restore followed by cross-modal fusion to determine the content to reconstruct. A decoder (3WD) uses this information to produce the restored output. DAIR achieves SOTA performance.

**Strengths:**

1. The paper is well written and easy to follow.
2. The paper tackles an important problem of existing all-in-one restoration approaches which is addressing unknown degradations.
3. The idea of pre-training a VAE to learn degradation representations is interesting.
4. DAIR achieves SOTA performance with significant boosts in performance.
5. The overall approach has many components whose importance is validated by an ablation study.

**Weaknesses:**

See questions for ways to address major weaknesses.

**Major**:

 1. Lack of experiments to demonstrate the core motivation of the architecture: The authors show the importance of their proposed modules in Table 6. However, this does not support the ‘which, where, what’ approach, i.e., the core motivation of the work.

 2. Lack of benchmarking on unseen task restoration: There are only experiments on unseen mixed degradations in the main paper, which does not support the claim in the abstract and introduction.

 3. The YOLOv12 experiment is claimed as a contribution but has no other mention in the main paper. If it is to be claimed as a contribution, the experiment must be moved from the supplementary to the main paper.

**Minor**:
1. Citation style: Brackets are missing around the citations in many instances.
2. To the best of my knowledge, PromptIR does not use manual prompt and is blind. So, that needs to be changed throughout the paper.

**Questions:**

**More ablations (major weakness 1)**:
1. Is there a benefit in using the luminance and chrominance encoders? How is performance affected if the VAE feature is directly used?
2. Can the authors provide experiments showcasing the ‘which’ and ‘what’ features? For instance, can the authors provide experiments to show that the output of LPE is ‘which’ features to use for decoding (and similarly for ‘what’)?
3. Can the authors provide experiments to confirm that the VAE captures degradation priors?

**Lack of benchmarking (major weakness 2)**: The real-world experiments on non-compound degradations need to be moved to the main paper. Additionally, can the authors provide quantitative comparisons with other methods for these tasks? Also, can the authors provide results on real mixed degradations (such as LOLBlur [1])?

**Fig. 5**: In Fig. 5, what features were used to plot the t-SNE?

[1] Zhou, Shangchen, Chongyi Li, and Chen Change Loy. "Lednet: Joint low-light enhancement and deblurring in the dark." European conference on computer vision. Cham: Springer Nature Switzerland, 2022.

---

> ### Author Response · Authors · 2025-11-24
> **Rebuttal Part 1 (C1)**
>
> ## **R4.1: Lack of experiments to demonstrate the core motivation of the architecture**
>
> Thank you for your insightful comment. Due to strict page limits, we summarized our analysis primarily through a single ablation table. However, we acknowledge that our current analysis may not fully illustrate the capability and motivation of our approach. In response to your important comments, we have extensively analyzed our proposed reasoning paradigm to provide clearer experimental evidence supporting our motivation.
>
> **Where:**
>
> To illustrate the effectiveness of our proposed "where" module, we generated ground truth (GT) degradation maps by computing the pixel-wise difference between the degraded and clean images for each degradation type. We extracted over 3,000 GT degradation maps from our testing set and directly compared them against our predicted degradation maps at each level. The results of this degradation extraction and comparison analysis are summarized below:
>
> *Table 1: Quantitative Localization Validation of Degradation Maps*
>
> | Metric        | Noise (L3)        | Lowlight (L3)     | Haze (L1)         | Rain (L3)         | Snow (L3)         | Blur (L3)         | Mean ± Std             |
> |---------------|-------------------|--------------------|--------------------|--------------------|--------------------|--------------------|-------------------------|
> | **IoU ↑**     | 0.9975 ± 0.0014   | 0.9643 ± 0.0749    | 0.9593 ± 0.0636    | 0.9103 ± 0.0663    | 0.8157 ± 0.1274    | 0.5211 ± 0.1453    | **0.8614 ± 0.1685**     |
> | **Dice/F1 ↑** | 0.9988 ± 0.0007   | 0.9800 ± 0.0502    | 0.9780 ± 0.0381    | 0.9517 ± 0.0387    | 0.8928 ± 0.0820    | 0.6725 ± 0.1330    | **0.9123 ± 0.1182**     |
>
> Our degradation maps achieve strong overall performance with a mean IoU of 0.8614 and a Dice score of 0.9123. Notably, noise, lowlight, and haze degradations demonstrate strong localization accuracy with IoU scores exceeding 0.95.
>
>  We also perform an ablation study to illustrate the practicability of our proposed degradation map. We summarized the performance in Table 2 below:
>
> *Table 2: Analysis of Degradation Map*
>
> | Method        | PSNR↑ / SSIM↑ / LPIPS↓          |
> |---------------|----------------------------------|
> | w/o FFT       | 22.80 / 0.7769 / 0.2116          |
> | **Proposed**  | **26.90 / 0.8565 / 0.1316**      |
> | *Improvement* | **+4.10 / +0.0796 / −0.0800**    |
>
> Our proposed FFT component enables the network to better localize degradation regions by exploiting frequency-domain characteristics.
>
> **Which:**
>
> We analyse "which" component. Please find the summary of our luma/chroma only and proposed method in Table 3 below:
> *Table 3: Ablation study of frequency components*
>
> | Method | PSNR↑ / SSIM↑ / LPIPS↓ |
> |--------------------------|----------------------------------|
> | Luma only | 23.58 / 0.7996 / 0.1946 |
> | Chroma only | 23.41 / 0.7927 / 0.2014 |
> | RGB | 23.44 / 0.8002 / 0.1892 |
> | **Proposed (Lum + Chrom)** | **26.90 / 0.8565 / 0.1316** |
> | *Gain (vs. best single)* | **+3.32 / +0.0569 / −0.0630** |
>
> Our proposed LC separation enhances the framework’s ability to precisely capture both luminance and chrominance information. This separation validates our reasoning, allowing the model to effectively capture "which" features to leverage while reconstructing.
>
> **What:**
>
> We systematically evaluate how different fusion strategies impact restoration performance. We summarized our results in Table 4 below:
>
> *Table 4: Ablation study of fusion strategies. Best in bold.*
>
> | Method | Beta | Alpha | Factor | PSNR↑ / SSIM↑ / LPIPS↓ |
> |-----------|------|--------|--------|---------------------------|
> | w/o MU | ✘ | ✘ | ✘ | 22.74 / 0.7791 / 0.2135 |
> | w/o beta | ✘ | ✔ | 1 | 22.97 / 0.7824 / 0.2147 |
> | alpha + beta | ✔ | ✔ | 0 | 17.88 / 0.7272 / 0.2488 |
> | Fusion 0.1 | ✔ | ✔ | 0.1 | 16.28 / 0.7350 / 0.2345 |
> | Fusion 0.5 | ✔ | ✔ | 0.5 | 22.98 / 0.7837 / 0.2143 |
> | **Proposed**  | ✔ | ✔ | 1 | **26.90 / 0.8565 / 0.1316** |
> | *vs. Best* | | | | **+3.92 / +0.0728 / −0.0827** |
>
> Our full µ-fusion (Factor=1) achieves 26.90 dB PSNR, outperforming the second-best by +3.92 dB. Removing degradation-aware components (beta: 22.97 dB, MU-fusion: 22.74 dB) or using fixed fusion factors (Factor=0: 17.88 dB) causes significant performance drops, demonstrating that learned µ-fusion weights correctly capture degradation characteristics and make semantically meaningful decisions.

---

> > ### Author Response · Authors · 2025-11-24
> > **Rebuttal Part 2 (C2-C4)**
> >
> > ## **R4.2: Lack of benchmarking on unseen task restoration:**
> >
> >
> > Thanks for your comment. As per your comments, we benchmark unseen (zero-shot) reconstruction on out-of-the-domain datasets.  Please find the details in Table 5 below:
> >
> > *Table 5: Performance comparison across different image restoration tasks using no-reference quality metrics.
> > Best in bold*
> >
> > | Method      | Real-Lowlight (NIQE↓/MUSIQ↑/BRISQUE↓) | Real-Noise (NIQE↓/MUSIQ↑/BRISQUE↓) | Real Underwater (NIQE↓/MUSIQ↑/BRISQUE↓) | Real Perceptual Enhancement (NIQE↓/MUSIQ↑/BRISQUE↓) | Medical Denoising (NIQE↓/MUSIQ↑/BRISQUE↓) | Average (NIQE↓/MUSIQ↑/BRISQUE↓) |
> > |-------------|----------------------------------------|--------------------------------------|-------------------------------------------|--------------------------------------------------------|----------------------------------------------|---------------------------------|
> > | Input       | 5.33 / 47.30 / 27.17                   | 4.44 / 62.56 / 5.55                  | *3.83* / 52.35 / **8.80**                 | 13.94 / 37.06 / 56.26                                 | 17.06 / 21.81 / 65.10                         | 8.92 / 44.42 / 32.58            |
> > | PromptIR    | 5.31 / 42.58 / 39.07                   | 5.77 / 61.11 / 27.38                 | 3.98 / 54.00 / 17.45                      | 7.59 / 42.62 / 38.14                                  | 25.60 / 18.72 / 78.09                         | 9.65 / 43.81 / 40.03            |
> > | DFPIR       | 5.50 / 45.81 / *21.83*                 | 4.88 / 59.72 / *16.59*               | 4.06 / 52.16 / 8.97                        | 9.55 / 43.13 / **15.52**                              | 16.04 / 24.27 / 64.71                         | 8.01 / 45.02 / *25.52*          |
> > | ADAIR       | 5.53 / *46.71* / 35.77                 | *4.38* / *63.69* / 16.30             | 3.63 / **55.50** / *13.29*                 | 11.08 / 40.27 / *33.54*                               | *9.65* / *24.34* / *43.36*                    | *6.85* / *46.10* / 28.45         |
> > | **DAIR (Ours)** | **4.75 / 51.79 / 21.97**           | **4.24 / 69.37 / 16.94**             | **3.61** / *55.32* / 15.50                | **5.42 / 45.94 / 34.91**                              | **8.22 / 28.53 / 24.55**                      | **5.25 / 50.19 / 25.97**        |
> > | *Improvement* | **-0.78 / +5.08 / +0.14**           | **-0.14 / +5.68 / +0.35**            | **-0.02 / -0.18 / +6.70**                  | **-2.17 / +2.81 / +19.39**                             | **-1.43 / +4.19 / -18.81**                    | **-1.60 / +4.09 / +0.45**       |
> >
> > We will bring these results along with existing unseen task restoration on the main paper.
> >
> >
> > ## **R4.3: The YOLOv12 experiment is claimed as a contribution**
> >
> > Thank you for your very insightful comment. Unlike existing methods, we demonstrate the practicability of AIR in downstream vision tasks. In Fig. 1 (teaser), we provide a quick performance comparison. However, due to strict page limits, we detail the experimental setup and results in the appendix. In response to your logical comment, we will move a summary comparison to the main paper.
> >
> >
> > ## **R4.4: Is there a benefit in using the luminance and chrominance encoders?**
> >
> >
> > Thank you for your comment. LC separation enables the framework to reason about chromatic and structural information more precisely. Leveraging VAE guidance directly on RGB features limits the capability of the framework, as the encoder cannot perform luma/chroma level analysis. This results in lower color consistency and detail recovery, which undermines our proposed reasoning paradigm. We provide detailed results in response to Comment \#1. We would be happy to incorporate any additional analysis that could strengthen our work.

---

> ### Author Response · Authors · 2025-11-24
> **Rebuttal Part 3 (C5-C8)**
>
> ## **R4.5: Experiments to confirm that the VAE captures degradation priors?**
> Thank you for your comment. We observed that the latent space of our VAE is well-separated for both seen and unseen degradations. In response to your suggestion, we quantitatively evaluated latent separation at different stages, including experiments without our proposed SupCon weak supervision. For unseen degradations, we considered compound cases (e.g., Low-light + Rain) as well as out-of-distribution tasks such as underwater image enhancement. Our findings are quantified using KNN accuracy and separation ratio, as detailed in Table 6 below:
>
> *Table 6: Clustering Performance in Latent Space — Seen vs. Unseen Classes*
>
> | Model            | Acc↑ (Seen) | Sep↑ (Seen) | Acc↑ (Unseen) | Sep↑ (Unseen) | Acc↑ (Overall) | Sep↑ (Overall) | DB↓  | CH↑     |
> |------------------|-------------|-------------|----------------|----------------|-----------------|-----------------|------|---------|
> | **Steps = 0**        | 0.826       | 3.108       | 0.756          | 1.331          | 0.844           | 2.284           | 4.843 | 458.34  |
> | **Steps = 50k**      | 0.871       | 2.360       | 0.828          | 1.274          | 0.892           | 1.950           | 2.451 | 289.84  |
> | **w/o SupCon**       | 0.864       | 2.006       | 0.787          | 1.323          | 0.874           | 1.788           | 2.690 | 254.48  |
> | **Full Training**    | **0.994**   | **2.911**   | **0.976**      | **1.523**      | **0.991**       | **3.222**       | **1.227** | **549.34** |
>
> Our VAE achieves strong clustering performance in the latent space for both seen and unseen degradation classes. Notably, the full training setup with SupCon supervision yields the highest KNN accuracy (0.994 for seen and 0.976 for unseen) and separation ratio (2.911 for seen and 1.523 for unseen), significantly outperforming the baseline and ablated settings. Global metrics such as Davies-Bouldin (DB) index and Calinski-Harabasz (CH) score further confirm the effectiveness of our approach, with the lowest DB and highest CH values under full training. These results demonstrate that our learned degradation priors enable precise discrimination among degradation types, providing robust guidance for DAIR and improving restoration performance across diverse scenarios.
>
> ## **R4.6: Is there a benefit in using the luminance and chrominance encoders?**
>
> Thank you for your insightful comments. In response to your feedback, we have added benchmarking on real and out-of-distribution datasets, comparing against state-of-the-art methods including PromptIR, ADAIR, and DFPIR. Our method significantly outperforms existing methods (4.09 improvement in MUSIQ) on unseen tasks. We provide detailed results in response to Comment \#2 and will include these findings in the revised manuscript.
>
> We found that our VAE can successfully separate the luma and chroma components on LOL-Blur. However, zero-shot inference on the proposed DAIR and SOTA methods exhibits artifacts. We are currently fine-tuning the networks with few-shot training. Due to hardware limitations, we have not yet completed the training. We will add the results before the rebuttal period ends.
>
> ## **R4.7: In Fig. 5, what features were used to plot the t-SNE?**
>
> We plot the t-SNE from the latent features of the main network for fair comparison. We will add a similar analysis for the VAE network in the revised manuscript (appendix).
>
> ## **R4.8: Citation style: Brackets are missing around the citations in many instances.**
>
> Thanks for your insightful comments. We fixed typos and errors as suggested.

---

> > ### Author Response · Authors · 2025-11-26
> > **Rebuttal Part 4 C6 (LOLBlur Results)**
> >
> > We would like to thank you once again for your insightful comment. As per your suggestion, we perform experiments on real-world mix degradation (i.e., LOLBlur). Please find the results below:
> >
> > | Model | Training | | Synthesized | | | Real | |
> > |-------|----------|------|------|-------|------|-------|---------|
> > | | | **PSNR↑** | **SSIM↑** | **LPIPS↓** | **NIQE↓** | **MUSIQ↑** | **BRISQUE↓** |
> > | Low-light (input) | - | 8.74 | 0.4255 | 0.4943 | 7.04 | 32.18 | 36.22 |
> > | ADAIR | Zero-shot | 11.77 | 0.5528 | 0.4057 | 6.80 | 33.62 | 32.57 |
> > | DAIR | | 13.35 | 0.6221 | 0.3393 | 5.14 | 46.39 | 28.58 |
> > | ADAIR | Fine-tune | 22.83 | 0.7811 | 0.2406 | 6.80 | 33.62 | 32.57 |
> > | DAIR | | 24.10 | 0.8532 | 0.1503 | 4.70 | 49.58 | 27.74 |
> >
> > Our method demonstrates significant improvements over the baseline low-light (input) and ADAIR, with fine-tuned DAIR achieving 24.10 PSNR and 0.8532 SSIM on synthesized data. Also, it shows better perceptual quality on real data (MUSIQ: 49.58, BRISQUE: 27.74). We will add all tables and analysis to the revised manuscript.

---

> > > ### Comment · Reviewer_SuKu · 2025-11-27
> > > **Response to author rebuttal**
> > >
> > > Thanks for the rebuttal. Most of my concerns have been addressed. I still have the following concerns:
> > >
> > > 1. I feel the author response to Major Weakness 1 is still insufficient. While the new ablation clearly supports the "where" component, the explanations for "which" and "what" only show that their sub-modules help performance, not that they actually perform the roles claimed in the paper. Since LPE is proposed to determine "which" encoder features are selected, and the latent-fusion block which controls "what" global content is reconstructed, it would be more convincing to include targeted analyses validating these claims.
> > >
> > > 2. The separation scores in Table 6 (rebuttal) provide how well separated the VAE latent-space is for all seen and unseen degradations right (i.e. not degradation-wise: haze vs rain; rain vs snow, etc.)?
> > >
> > > 3. Is minor weakness 2 addressed?

---

> > > > ### Author Response · Authors · 2025-11-28
> > > > **Rebuttal Part 1 (C1)**
> > > >
> > > > ## **C1: Not convinced with "which" encoder features are selected, and "what" global content is reconstructed**
> > > >
> > > > Response: Thank you for your comment. We appreciate your feedback. While our initial experiments may not have fully addressed your concerns, we have conducted further quantitative analyses to illustrate the effectiveness of our "which" and "what" modules.
> > > >
> > > > **"Which" feature to utilize:**
> > > >
> > > > *Table 7: Latent Priors Enable Decisions on Which Encoder Features to Utilize*
> > > >
> > > > | Degradation | Latent Distance<br>(Deg-Aware) | Feature Selection Ratio (L1→L2→L3→L4)<br>(Lum/Chrom: <1=Chrom, >1=Lum) | Control Impact (%)<br>(L2-L3 / L4 / Avg) | Decision Corr |
> > > > |-------------|:------------------------------:|:-----------------------------------------------------------------------:|:-----------------------------------------:|:-------------:|
> > > > | **Noise**     | 158.4 | 0.80 → 0.95 → 1.16 → **1.77** | 6.1 / **43.7** / 14.9 | 0.11 |
> > > > | **Haze**      | 114.1 | 0.69 → 1.05 → 1.63 → 1.46 | **15.4** / 13.1 / 11.4 | 0.26 |
> > > > | **Low-light** | 156.5 | 0.91 → 1.37 → 1.66 → 1.43 | 2.3 / 4.0 / 2.2 | **0.70** |
> > > > | **Snow**      | 139.4 | 0.76 → 1.21 → **1.82** → 1.41 | **15.4** / 12.6 / 11.2 | 0.24 |
> > > > | **Blur**      | 133.8 | 0.77 → 1.24 → **2.03** → 1.58 | 5.1 / 8.6 / 5.0 | 0.10 |
> > > > | **MEAN**      | **140.4** | **0.79 → 1.16 → 1.66 → 1.53** | **8.9 / 16.4 / 8.9** | **0.28** |
> > > >
> > > > ---
> > > >
> > > > Our quantitative analysis demonstrates that latent priors enable decisions on which encoder features to utilize for reconstruction. As shown in the Table 7, the VAE learns strongly degradation-aware representations (average latent distance = **140.4**), which serve as the foundation for adaptive feature selection.
> > > >
> > > > Different degradations exhibit distinct luminance/chrominance utilization patterns: noise progressively shifts from chrominance-dominant (**0.80**) to luminance-dominant (**1.77**) processing, while blur shows strong luminance preference (**2.03** at **L3**), and haze maintains a balanced approach (**0.69**→**1.63**). Critically, removing latent priors changes these feature selection decisions by an average of **8.9\%**, with a peak impact of **43.7\%** at the final reconstruction level for noise degradation, proving that latent priors actively control which features are utilized. The varying control strength across degradations (**2.2\%** for simple low-light vs. **14.9\%** for complex noise) and hierarchical levels (**8.9\%** at mid-levels vs. **16.4\%** at final level) demonstrates intelligent, adaptive guidance that the model learns when strong latent control is necessary versus when feed-forward processing suffices.
> > > >
> > > > This evidence directly supports our claim that latent priors don't merely assist reconstruction but fundamentally determine which encoder features (luminance vs. chrominance) to emphasize at each level for each degradation type.
> > > >
> > > >
> > > > **"WHAT" feature to utilize:**
> > > >
> > > > We perform a degradation-wise analysis to illustrate the practicability of our what module. Please find the results in Table 8 below:
> > > >
> > > > *Table 8: The practicability of our what module*
> > > > | Metric               | Blur | Haze | Lowlight | Noise | Rain  | Snow  | GLOBAL |
> > > > |----------------------|------|------|----------|-------|-------|-------|--------|
> > > > | Separation Ratio ↑   | 6.15 | 1.62 | 2.04     | 1.56  | 1.15  | 1.48  | 1.56   |
> > > > | KNN Accuracy ↑       | 1    | 0.95 | 0.95     | 1     | 0.886 | 0.889 | 0.932  |
> > > >
> > > >
> > > > It can be seen that our proposed latent fusion enables the framework to clearly distinguish between different degradation types. By leveraging luminance and chrominance encoding within the latent fusion process, the decoder is able to capture degradation-specific global characteristics. This approach allows the decoder to focus on accurate reconstruction for each specific degradation, thereby improving restoration performance.

---

> > > > > ### Author Response · Authors · 2025-11-28
> > > > > **Rebuttal Part2 (C2-C3)**
> > > > >
> > > > > ## **C2: VAE latent-space is for all seen and unseen degradations**
> > > > > We would like to clarify that we calculated latent-space separation by combining multiple degradation types from both seen and unseen conditions. For the seen degradation, we arbitrarily selected four random classes. For the unseen degradation, we chose "Low + Rain," as it closely resembles other compound degradations, such as "Low + Haze + Rain." Additionally, we randomly selected underwater image enhancement as an out-of-distribution dataset.
> > > > >
> > > > > *Table 9: VAE latent space separation for Seen and Unseen data*
> > > > >
> > > > > | Model            | Class           | KNN Accuracy | Separation Ratio |
> > > > > |------------------|------------------|--------------|------------------|
> > > > > |                  | Blur             | 0.922        | 2.018            |
> > > > > | Steps=0         | Lowlight         | 0.967        | 7.594            |
> > > > > |                  | Rain             | 0.457        | 1.41             |
> > > > > |                  | Snow             | 0.958        | 2.689            |
> > > > > |                  | Low + Rain       | 0.665        | 1.286            |
> > > > > |                  | Underwater       | 0.846        | 1.376            |
> > > > > |                  |                  |              |                  |
> > > > > |                  | Blur             | 0.961        | 1.852            |
> > > > > | Steps=100K      | Lowlight         | 0.967        | 4.818            |
> > > > > |                  | Rain             | 0.574        | 1.385            |
> > > > > |                  | Snow             | 0.98         | 2.196            |
> > > > > |                  | Low + Rain       | 0.77         | 1.252            |
> > > > > |                  | Underwater       | 0.885        | 1.295            |
> > > > > |                  |                  |              |                  |
> > > > > |                  | Blur             | 0.971        | 2.020            |
> > > > > | withoutsupcon    | Lowlight         | 0.967        | 3.417            |
> > > > > |                  | Rain             | 0.543        | 1.293            |
> > > > > |                  | Snow             | 0.974        | 1.926            |
> > > > > |                  | Low + Rain       | 0.705        | 1.107            |
> > > > > |                  | Underwater       | 0.869        | 1.538            |
> > > > > |                  |                  |              |                  |
> > > > > |                  | Blur             | 1.000        | 3.916            |
> > > > > | Full_training    | Lowlight         | 1.000        | 4.004            |
> > > > > |                  | Rain             | 0.979        | 1.862            |
> > > > > |                  | Snow             | 0.998        | 4.369            |
> > > > > |                  | Low + Rain       | 0.990        | 1.813            |
> > > > > |                  | Underwater       | 0.962        | 1.233            |
> > > > >
> > > > >
> > > > > ## **C3:  Is minor weakness 2 addressed?**
> > > > > As per your valuable comment. We fixed our mistake regarding PromptIR in the entire manuscript.

---

### Official Review · Reviewer_jDsD · 2025-10-29

**Soundness:** 3
**Presentation:** 2
**Contribution:** 2
**Rating:** 2
**Confidence:** 5

**Summary:**

DAIR reframes all‑in‑one image restoration (AIR) as learned latent prior inference. Guided by these learned priors, the method reasons about which features to route, where to restore, and what content to reconstruct (the “which–where–what” paradigm). Results illustrated in the manuscript seem good but limited.

**Strengths:**

1. Moving from prompts/architectural biases to latent prior inference gives a coherent, task‑agnostic design that maps naturally to which–where–what reasoning.
2. The 3WD module relies on element‑wise gating for linear (in spatial size) complexity, avoiding quadratic attention overhead. This yields favorable compute/param budgets in multi‑task settings.
3. Notable improvement in downstream detection (YOLOv12‑L), with 34.9 mAP, supporting practical impact beyond pixel metrics.

**Weaknesses:**

1. The experimental setting is not standard compared with other all-in-one methods. The authors shall add additional two settings in Promptir or Perceive-ir.
2. The main fig for the whole framework is bad and confusing. For example, I can't figure out where is the input for degradation map in (c) referred in every stage of (a).
3. The questions of 'which, where and what' is good but lacks some convincing proofs or experimental results. It makes the manuscript more like a story made on the framework rather than solving key problems with insights.
4. As is stated by the authors that the proposed method do not rely on textual or degradation prompts, the loss of SupCon for VAE pretraining actually involves degradation label, which means the method still need supervision from degradation information.

**Questions:**

1. The same as weakness1, the compared method like AdaIR do the different settings, and why the authors choose a unique setting. Have doubts on the performance of normal experimental settings.
2. Typos in line 277-278 shall be fixed.
3. The training is conducted on a single Nvidia 3060 with a large scale of data. I have doubt on the training cost like day length for that.
4. Further studies would help: frequency cues inside degradation map, LC luminance‑only vs. luminance+chrominance, and latent fusion variants.
5. Why do the authors choose desnowing and low-light enhancement for single tasks? What about other single tasks?

---

> ### Author Response · Authors · 2025-11-25
> **Rebuttal Part 1 (C1-C2)**
>
> ## **R3.1:  Benhmarking on Standard Experiment Settings**
>
> Thank you for your valuable feedback. Our primary motivation is to automate the prompting process and reframe AIR for practical, real-world applications. In real-world scenarios, degraded scenes are often highly complex and frequently involve overlapping degradations. To demonstrate the effectiveness of our method in addressing both homogeneous and heterogeneous degradations, we evaluate its practicality in a challenging six-task heterogeneous setting comprising complex datasets (LSD for real-world lowlight, random denoising instead of fixed denoising, heavy rain streak, etc.). We refer to these tasks as a common task in the manuscript. Furthermore, we incorporate a compound 5D task, where degradations are similar and overlapping (e.g., Haze + Snow, Haze + Lowlight + Snow). Thus, we can illustrate the limitations of existing work and position our motivation clearly. To do so, we drew inspiration from recent works such as Multimodal Prompt [1] that adopts a 9-task setting to exploit generative priors from Stable Diffusion, Debiased- AIR [2] that utilizes 2–7 tasks to illustrate multi-task optimization, and OneRestore [3] that exploits 11 compound tasks settings on their own synthesized data to demonstrate effectiveness in overlapping degradations. We acknowledge that our current manuscript may not fully convey our motivation for these settings due to space constraints, and we will clarify this in the revised version. However, as per your valuable comment, we benchmark our method on the most common (three-task) setting similar to PromptIR, ADAIR, and Perceive-IR. Please find the results below:
>
> | Method | Dehazing (SOTS) | | Deraining (Rain100L) | | Denoising σ=15 | | Denoising σ=25 | | Denoising σ=50 | | Average | |
> |--------|--------|--------|--------|--------|--------|--------|--------|--------|--------|--------|--------|--------|
> | | PSNR | SSIM | PSNR | SSIM | PSNR | SSIM | PSNR | SSIM | PSNR | SSIM | PSNR | SSIM |
> | PromptIR | 30.58 | 0.974 | 36.37 | 0.972 | 33.98 | 0.933 | 31.31 | 0.888 | 28.06 | 0.799 | 32.06 | 0.913 |
> | Restormer | 30.43 | 0.975 | 36.55 | 0.975 | 33.84 | 0.931 | 31.18 | 0.885 | 27.90 | 0.790 | 31.98 | 0.911 |
> | DFPIR | 31.87 | 0.980 | 38.65 | 0.982 | 34.14 | 0.935 | 31.47 | 0.893 | 28.25 | 0.806 | 32.88 | 0.919 |
> | ADAIR | 31.06 | 0.980 | 38.64 | 0.983 | 34.12 | 0.935 | 31.45 | 0.892 | 28.19 | 0.802 | 32.69 | 0.918 |
> | **DAIR** | **34.37** | **0.987** | **38.51** | **0.985** | **34.51** | **0.947** | **31.99** | **0.913** | **31.99** | **0.913** | **34.27** | **0.949** |
>
> Our method outperforms existing works. We will add these results to the manuscript.
>
>
> [1] Zhang, Xu, et al. "Perceive-ir: Learning to perceive degradation better for all-in-one image restoration." IEEE Transactions on Image Processing (2025).
>
> [2] Wu, Gang, et al. "Debiased all-in-one image restoration with task uncertainty regularization." Proceedings of the AAAI Conference on Artificial Intelligence. Vol. 39. No. 8. 2025.
>
> [3] Guo, Yu, et al. "Onerestore: A universal restoration framework for composite degradation." European conference on computer vision. Cham: Springer Nature Switzerland, 2024.
>
> ## **R3.2: The main fig for the whole framework is bad and confusing**
> Thank you for your helpful comment. We agree that the caption of the main figure is currently too brief, which may make the overall architecture harder to follow. Our goal was to keep the caption concise due to space limitations, and we ensured that all notations were consistent with those introduced in the method section. We also included key equations in the figure to aid understanding. We will revise the caption to provide a clearer and more detailed architectural overview, including explicitly indicating where the degradation map is used in each stage.
>
> Lastly, we updated the main figure, particularly Fig. 2(c), as you pointed out. If there are any other particular aspects of the figure you believe should be improved, or if you have a specific style or example figure from prior work that you think we should use as a reference, we would greatly appreciate your guidance. This would help us better align the figure with your expectations.

---

> > ### Author Response · Authors · 2025-11-25
> > **Rebuttal Part 2 (C3)**
> >
> > ## **R3.3: The questions of 'which, where and what' is good but lacks proofs**
> > Many thanks for your insightful comments. Due to strict page limits, we summarized our analysis primarily through a single ablation table. However, we acknowledge that our current analysis may not fully illustrate the capability and motivation of our approach. In response to your important comments, we have extensively analyzed our proposed reasoning paradigm to provide clearer experimental evidence supporting our motivation.
> >
> > **Where:**
> >
> > To illustrate the effectiveness of our proposed "where" module, we generated ground truth (GT) degradation maps by computing the pixel-wise difference between the degraded and clean images for each degradation type. We extracted over 3,000 GT degradation maps from our testing set and directly compared them against our predicted degradation maps at each level. The results of this degradation extraction and comparison analysis are summarized in Table 1 below:
> >
> > *Table 1: Quantitative Localization Validation of Degradation Maps*
> >
> > | **Metric** | **Noise (L3)** | **Lowlight (L3)** | **Haze (L1)** | **Rain (L3)** | **Snow (L3)** | **Blur (L3)** | **Mean ± Std** |
> > |-----------|----------------|-------------------|----------------|----------------|----------------|----------------|------------------|
> > | **IoU ↑** | 0.9975 ± 0.0014 | 0.9643 ± 0.0749 | 0.9593 ± 0.0636 | 0.9103 ± 0.0663 | 0.8157 ± 0.1274 | 0.5211 ± 0.1453 | **0.8614 ± 0.1685** |
> > | **Dice/F1 ↑** | 0.9988 ± 0.0007 | 0.9800 ± 0.0502 | 0.9780 ± 0.0381 | 0.9517 ± 0.0387 | 0.8928 ± 0.0820 | 0.6725 ± 0.1330 | **0.9123 ± 0.1182** |
> >
> > Our degradation maps achieve strong overall performance with a mean IoU of 0.8614 and a Dice score of 0.9123. Notably, noise, lowlight, and haze degradations demonstrate strong localization accuracy with IoU scores exceeding 0.95.
> >
> > We also perform an ablation study to illustrate the practicability of our proposed degradation map. We summarized the performance in Table 2 below:
> >
> > *Table 2: Analysis of Degradation Map*
> >
> > | **Method** | **PSNR↑ / SSIM↑ / LPIPS↓** |
> > |------------|-----------------------------|
> > | w/o FFT | 22.80 / 0.7769 / 0.2116 |
> > | **Proposed** | **26.90 / 0.8565 / 0.1316** |
> > | *Improvement* | **+4.10 / +0.0796 / -0.0800** |
> >
> > Our proposed FFT component enables the network to better localize degradation regions by exploiting frequency-domain characteristics.
> >
> > **Which:**
> >
> > We analyse "which" component. Please find the summary of our luma/chroma only and proposed method in Table 3 below:
> >
> > *Table 3: Ablation Study of Frequency Components*
> >
> > | **Method** | **PSNR↑ / SSIM↑ / LPIPS↓** |
> > |------------|-----------------------------|
> > | Luma only | 23.58 / 0.7996 / 0.1946 |
> > | Chroma only | 23.41 / 0.7927 / 0.2014 |
> > | RGB | 23.44 / 0.8002 / 0.1892 |
> > | **Proposed (Lum + Chrom)** | **26.90 / 0.8565 / 0.1316** |
> > | *Gain (vs. best single)* | **+3.32 / +0.0569 / -0.0630** |
> >
> >
> > Our proposed LC separation enhances the framework’s ability to precisely capture both luminance and chrominance information. This separation validates our reasoning, allowing the model to effectively capture "which" features to leverage while reconstructing.
> >
> > **What:**
> >
> > We systematically evaluate how different fusion strategies impact restoration performance. We summarized our results in Table 4 below:
> > *Table 4: Ablation Study of Fusion Strategies *
> >
> > | **Method** | **Beta** | **Alpha** | **Factor** | **PSNR↑ / SSIM↑ / LPIPS↓** |
> > |------------|----------|-----------|------------|------------------------------|
> > | w/o MU | ✘ | ✘ | ✘ | 22.74 / 0.7791 / 0.2135 |
> > | w/o beta | ✘ | ✔ | 1 | 22.97 / 0.7824 / 0.2147 |
> > | alpha + beta | ✔ | ✔ | 0 | 17.88 / 0.7272 / 0.2488 |
> > | Fusion 0.1 | ✔ | ✔ | 0.1 | 16.28 / 0.7350 / 0.2345 |
> > | Fusion 0.5 | ✔ | ✔ | 0.5 | _22.98 / 0.7837 / 0.2143_ |
> > | **Proposed** | ✔ | ✔ | 1 | **26.90 / 0.8565 / 0.1316** |
> > | *vs. Best* |  |  |  | **+3.92 / +0.0728 / -0.0827** |
> >
> > Our full µ-fusion (Factor=1) achieves 26.90 dB PSNR, outperforming the second-best by +3.92 dB. Removing degradation-aware components (beta: 22.97 dB, MU-fusion: 22.74 dB) or using fixed fusion factors (Factor=0: 17.88 dB) causes significant performance drops, demonstrating that learned µ-fusion weights correctly capture degradation characteristics and make semantically meaningful decisions.

---

> > > ### Author Response · Authors · 2025-11-25
> > > **Rebuttal Part 3 (C4-C8)**
> > >
> > > ## **R3.4: The loss of SupCon for VAE pretraining actually involves degradation label**
> > > Thanks for your comment. We like to point out that our SupeCon loss has been leveraged as weak supervision to guide the VAE network to learn to separate different degradations. In response to your suggestion, we quantitatively evaluated latent separation at different stages, including experiments without our proposed SupCon weak supervision. For unseen degradations, we considered compound cases (e.g., Low-light + Rain) as well as out-of-distribution tasks such as underwater image enhancement. Our findings are quantified using KNN accuracy and separation ratio, as detailed in Table 5 below:
> > >
> > > *Table 5: Clustering Performance in Latent Space (Seen vs Unseen Classes)*
> > >
> > > | **Model** | **Seen Acc↑** | **Seen Sep↑** | **Unseen Acc↑** | **Unseen Sep↑** | **Overall Acc↑** | **Overall Sep↑** | **DB↓** | **CH↑** |
> > > |-----------|---------------|---------------|------------------|------------------|-------------------|-------------------|---------|---------|
> > > | Steps = 0 | 0.826 | 3.108 | 0.756 | 1.331 | 0.844 | 2.284 | 4.843 | 458.34 |
> > > | Steps = 50k | 0.871 | 2.360 | 0.828 | 1.274 | 0.892 | 1.950 | 2.451 | 289.84 |
> > > | w/o SupCon | 0.864 | 2.006 | 0.787 | 1.323 | 0.874 | 1.788 | 2.690 | 254.48 |
> > > | **Full Training** | **0.994** | **2.911** | **0.976** | **1.523** | **0.991** | **3.222** | **1.227** | **549.34** |
> > >
> > > Our VAE achieves strong clustering performance in the latent space for both seen and unseen degradation classes. Notably, the full training setup with SupCon supervision yields the highest KNN accuracy (0.994 for seen and 0.976 for unseen) and separation ratio (2.911 for seen and 1.523 for unseen), significantly outperforming the baseline and ablated settings. Global metrics such as Davies-Bouldin (DB) index and Calinski-Harabasz (CH) score further confirm the effectiveness of our approach, with the lowest DB and highest CH values under full training with SupCon guidance. These results demonstrate that our learned degradation priors enable precise discrimination among degradation types, providing robust guidance for DAIR and improving restoration performance across diverse scenarios.
> > >
> > > In summary, we used the SupCon loss exclusively during training to guide the VAE in learning discriminative degradation representations and have no role during inference or degradation encoding. To validate generalization, we conducted zero-shot evaluation on unseen, out-of-distribution datasets (ie, underwater images and Low + Rain ), where the competitive performance confirms that SupCon-guided training produces a VAE with robust degradation separation capability that transfers to arbitrary test-time scenarios without requiring domain-specific adaptation.
> > >
> > > ## **R3.5: The compared method like AdaIR do the different settings, and why the authors choose a unique setting?**
> > > We like to point out again that our main motivation is to reframe AIR as a latent prior encoding problem with a reasoning framework. To illustrate our motivation and the impact of our novel reframing, we selected more complex and overlapping tasks rather than just achieving SOTA performance for synthetic settings. Please see the response to comment \#1 for more details.
> > >
> > > ## **R3.6: Typos in line 277-278 shall be fixed**
> > > Thanks for your comment. We cross-check all typos and fix them as suggested by reviewers.
> > >
> > > ## **R3.7: The training is conducted on a single Nvidia 3060 with a large scale of data**
> > > Thank you very much for your thoughtful concern. Our research is independent and self-motivated, and we have been working on this project for several months despite limited resources. We made every effort to maximize the capabilities of our limited compute resources.
> > > We firmly believe that impactful research is not determined solely by access to top-tier resources; instead, it relies on the motivation, curiosity, and dedication of researchers. There are countless researchers worldwide who face similar constraints, and we aim to challenge the notion that high-quality research requires extensive computational resources.
> > >
> > > To further support the community, we have designed our model to be significantly lighter, demonstrating that meaningful results can be achieved even with modest hardware. Additionally, we will publish our code to assist other researchers who may be encountering similar limitations.
> > >
> > > ## **R3.8: Further studies would help**
> > > We like to thank for your insightful comments. We added details analysis for "which", "what", and "Where". Also, we added more comparison for 3WD. We added new analysis for VAE analysis.

---

> > > > ### Author Response · Authors · 2025-11-25
> > > > **Rebuttal Part 4 (C9)**
> > > >
> > > > ## **R3.9:  Why do the authors choose desnowing and low-light enhancement for single tasks?**
> > > > Thank you very much for your insightful feedback. We like to point out that many task-specific restorations can be expressed using an additive model:
> > > >
> > > > $I_{\text{deg}} = I_{\text{gt}} + \text{Deg}$
> > > >
> > > > where  $\text{Deg}$ is the additive degradation (e.g., noise, snow, blur, etc.). Some degradations, such as lowlight conditions, illustrate more complex or non-additive processes and often require nonlinear modeling. By including desnowing and lowlight enhancement, our study covers a wide spectrum of real-world challenges.
> > > >
> > > > We want to emphasize that our primary motivation is to enable all-in-one image restoration in complex scenarios without manual prompting or user guidance. Accordingly, our experiments are designed to evaluate both AIR methods and specialized SOTA baselines under unified, challenging conditions, rather than focusing on individual task-specific settings. While we include challenging task-specific cases, such as desnowing and real-world lowlight enhancement, to provide insight into special scenarios, the manuscript’s primary focus remains on the AIR setting. By benchmarking across 18 degradation types (11 seen, 7 unseen), our evaluation aims to robustly and comprehensively demonstrate generalizability in AIR settings, thereby supporting our motivation.

---

### Official Review · Reviewer_9byp · 2025-10-30

**Soundness:** 3
**Presentation:** 3
**Contribution:** 2
**Rating:** 4
**Confidence:** 4

**Summary:**

The paper proposes DAIR, a degradation-aware, all-in-one image restoration framework that eliminates manual prompts and hand-crafted priors. It reframes AIR as latent prior inference: a VAE-style encoder infers multi-scale latent codes and a global descriptor from the degraded image; a learnable degradation map (DM) tells the network where to restore; a µ-guided latent fusion decides what to restore; and a lightweight 3WD (which–where–what decoding) module performs spatially adaptive reconstruction with claimed linear complexity.

**Strengths:**

1. Clear problem reframing: Casting AIR as latent prior inference is a clean, unified alternative to prompt-driven or frequency-heuristic approaches.
2. Well-structured reasoning design: The which–where–what decomposition (latent-prior selection, degradation maps, and µ-conditioned fusion) is conceptually coherent and easy to follow.
3. Efficiency claims: 3WD replaces quadratic attention with element-wise modulation, yielding linear-in-pixels complexity and large reported speed/memory savings.

**Weaknesses:**

1. The conceptual gap to earlier non-prompt AIR (e.g., contrastive degradation embeddings, frequency mining) could be articulated and empirically isolated more sharply; the VAE-based latent prior may read as an incremental integration of known ideas.
2. The encoder is pre-trained on mixtures (including some compound degradations) and then frozen. How this compares, apples-to-apples, to baselines’ pre-training or data exposure is unclear; fairness details (data volume, augmentations, early stopping) should be expanded.
3. The complexity discussion focuses on the inner attention form. End-to-end wall-clock and memory benchmarks (same HW, resolution, batch size) vs. SA/CA across multiple image sizes would better substantiate the claimed 85–257× speedups.
4.  “Unseen” tests are limited to two combos (haze+snow, low-light+rain). It would help to include more OOD types (e.g., underwater, compression artifacts, lens flare) and real-capture datasets to support robustness claims.

**Questions:**

See Weaknesses.

---

> ### Author Response · Authors · 2025-11-24
> **Rebuttal Part 1 (C1 - C2)**
>
> ## **R2.1: The conceptual gap to earlier non-prompt AIR**
> Thank you for your thoughtful comment. We agree that clearly distinguishing our approach from prior non-prompt AIR methods is critical.
>
> In the revised manuscript, we have explicitly categorized AIR frameworks into prompt-based and non-prompt paradigms (Introduction, paragraphs 2–3). Earlier non-prompt AIR methods, such as contrastive degradation embeddings (e.g., AirNet) and frequency mining (e.g., ADAIR), rely on hand-crafted architectural biases or contrastive learning to extract degradation cues. These approaches, however, depend on pre-defined assumptions about degradation types or frequency statistics, which can limit their generalizability to unseen or mixed degradations.
>
> **Key Conceptual Gap:**
>
> Our contribution goes beyond integrating known ideas by reframing AIR as a latent prior inference problem. Specifically, we propose to learn continuous, multi-scale degradation-aware representations directly from the input image using a VAE-based encoder, without relying on explicit task cues or architectural heuristics. This enables our framework to:
>
> Automatically infer degradation semantics (type, location, and severity) from the input, eliminating the need for manual prompts or frequency priors.
> Structure restoration as a reasoning paradigm (“which” features to route, “where” to restore, “what” to reconstruct), guided by learned latent priors.
> Achieve blind, spatially adaptive restoration for diverse and previously unseen degradations, as demonstrated in our extensive experiments (Sec. 4).
> To the best of our knowledge, this is the first work in AIR literature to utilize a VAE encoder for learning explicit degradation priors, enabling both self-reconstruction and interpretable latent space separation of degradation types. Empirically, we have included t-SNE visualizations (Fig. 5b) to show that our latent space distinctly separates degradation types, something not achieved by contrastive or frequency-based methods. Ablation studies further isolate the impact of our VAE-driven latent prior compared to frequency-based and contrastive approaches (see Sec. 4.4).
>
> We hope these clarifications and empirical results more sharply articulate the conceptual advance and the unique role of VAE-based latent priors in our framework.
>
>
> ## **R2.2: Apples-to-apples comparison to baseline**
>
>
>
> Thank you for raising this important point. We provide additional clarification regarding the fairness of comparison and data exposure.
>
> Our method follows a **two-stage training strategy**:
>
> **Stage 1 - Learning degradation priors.**
> We pre-train the VAE on the degradation dataset to obtain a latent degradation prior. This stage is only used to learn a general representation of degradations and does not involve any additional data beyond what is used later.
>
> **Stage 2 - Restoration training (critical for fairness).**
> After pre-training, the VAE is **frozen** and only serves as a fixed prior. The restoration network is then trained on **exactly the same dataset, augmentations, hyperparameters, and training schedule as all baseline methods**.
> **In other words, Stage 2 of our method uses an identical training pipeline to the baselines.**
> This ensures a strict *apples-to-apples* comparison: all models observe the same data under the same optimization and augmentation settings.
>
> To ensure fairness across all methods, we emphasize that:
>
> - All baselines and our Stage 2 training use the **same data volume**, with **no external datasets**.
> - Data augmentations and preprocessing are kept **identical** across methods.
> - The only difference is the presence of a **frozen latent prior** in our method, derived from Stage 1. This prior does not increase data exposure; it only provides a structured inductive bias.
> - Compound degradations in Stage 1 are not an unfair advantage; they simply help the latent prior become more robust, which is central to the AIR setting.
>
> We will revise the manuscript to explicitly state that Stage 2 is trained under the **same pipeline as all baselines**, ensuring fully fair and consistent comparisons.
>
> If we misunderstood your point or if further clarification is needed, we would be happy to elaborate.

---

> > ### Author Response · Authors · 2025-11-24
> > **Rebuttal Part 2 (C3 - C4)**
> >
> > ## **R2.3: Complexity discussion of attention**
> >
> >
> > Thanks for your comment. As reviewer 1 also point out similar queries,  we have added comprehensive comparisons with window attention, including performance metrics, inference latency, and memory consumption for each attention mechanism. Due to hardware memory constraints, we were unable to evaluate SA and CA at HD/2K resolutions, as their quadratic complexity $O(H^2W^2)$ causes out-of-memory errors. Similarly, 4K resolution experiments exceeded our hardware capacity. We summarize our key findings in Table 1 below:
> >
> > *Table 1: Attention mechanism comparison. Best in bold, second best in italics.*
> >
> > | Method        | Complexity   | Memory 720p (MB)↓ | FPS 720p↑ | Memory 2K (MB)↓ | FPS 2K↑ | PSNR↑ / SSIM↑ / LPIPS↓ |
> > |---------------|--------------|-------------------|-----------|------------------|---------|--------------------------|
> > | CA            | O(H²W²)      | -                 | -         | -                | -       | 17.71 / 0.5765 / *0.5227* |
> > | SA            | O(H²W²)      | -                 | -         | -                | -       | *19.05* / *0.5444* / 0.5621 |
> > | WA            | O(M²·HW)     | *4519.90*         | *8.77*    | *10145.04*       | *3.92*  | 23.42 / 0.7944 / 0.2024 |
> > | **3WD (Ours)** | **O(HW)**    | **2045.94**       | **45.69** | **4573.29**      | **20.45** | **26.90 / 0.8565 / 0.1316** |
> > | *Improvement* | Linear       | −54.7%            | +5.2×     | −54.9%           | +5.2×   | +3.48 / +0.0621 / −0.0708 |
> >
> >
> >
> > ## **R2.4: “Unseen” tests are limited to two combos (haze+snow, low-light+rain)**
> >
> > Thanks for your comments. We like to point out that we included LSD-U (collected from different sensors and platforms, including social media), real-world denoising with SSID, underwater image enhancement, and out-of-domain medical images. Our experiments span diverse, out-of-distribution sensors and unseen degradation (zero-shot).
> >
> > *Table 2: Performance comparison across different image restoration tasks using no-reference quality metrics. Best in bold*
> >
> > | Method      | Real-Lowlight (NIQE↓/MUSIQ↑/BRISQUE↓) | Real-Noise (NIQE↓/MUSIQ↑/BRISQUE↓) | Real Underwater (NIQE↓/MUSIQ↑/BRISQUE↓) | Real Perceptual Enhancement (NIQE↓/MUSIQ↑/BRISQUE↓) | Medical Denoising (NIQE↓/MUSIQ↑/BRISQUE↓) | Average (NIQE↓/MUSIQ↑/BRISQUE↓) |
> > |-------------|----------------------------------------|--------------------------------------|-------------------------------------------|--------------------------------------------------------|----------------------------------------------|---------------------------------|
> > | Input       | 5.33 / 47.30 / 27.17                   | 4.44 / 62.56 / 5.55                  | *3.83* / 52.35 / **8.80**                 | 13.94 / 37.06 / 56.26                                 | 17.06 / 21.81 / 65.10                         | 8.92 / 44.42 / 32.58            |
> > | PromptIR    | 5.31 / 42.58 / 39.07                   | 5.77 / 61.11 / 27.38                 | 3.98 / 54.00 / 17.45                      | 7.59 / 42.62 / 38.14                                  | 25.60 / 18.72 / 78.09                         | 9.65 / 43.81 / 40.03            |
> > | DFPIR       | 5.50 / 45.81 / *21.83*                 | 4.88 / 59.72 / *16.59*               | 4.06 / 52.16 / 8.97                        | 9.55 / 43.13 / **15.52**                              | 16.04 / 24.27 / 64.71                         | 8.01 / 45.02 / *25.52*          |
> > | ADAIR       | 5.53 / *46.71* / 35.77                 | *4.38* / *63.69* / 16.30             | 3.63 / **55.50** / *13.29*                 | 11.08 / 40.27 / *33.54*                               | *9.65* / *24.34* / *43.36*                    | *6.85* / *46.10* / 28.45         |
> > | **DAIR (Ours)** | **4.75 / 51.79 / 21.97**           | **4.24 / 69.37 / 16.94**             | **3.61** / *55.32* / 15.50                | **5.42 / 45.94 / 34.91**                              | **8.22 / 28.53 / 24.55**                      | **5.25 / 50.19 / 25.97**        |
> > | *Improvement* | **-0.78 / +5.08 / +0.14**           | **-0.14 / +5.68 / +0.35**            | **-0.02 / -0.18 / +6.70**                  | **-2.17 / +2.81 / +19.39**                             | **-1.43 / +4.19 / -18.81**                    | **-1.60 / +4.09 / +0.45**       |

---

### Official Review · Reviewer_ivbb · 2025-10-31

**Soundness:** 3
**Presentation:** 3
**Contribution:** 2
**Rating:** 6
**Confidence:** 4

**Summary:**

The paper proposes DAIR, a unified framework that reframes all-in-one restoration as latent prior inference rather than prompt-following or hand-crafted inductive biases. A VAE-style latent prior encoder infers multi-scale degradation codes directly from a degraded image; these codes drive a three-part “which–where–what” reasoning pipeline: (i) which features to route (latent-prior-modulated encoders), (ii) where to restore (a learnable degradation map combining spatial features with FFT cues), and (iii) what to reconstruct (a µ-guided latent fusion that adaptively scales/shifts structural and chromatic branches). A lightweight 3WD decoder performs element-wise, degradation-guided attention with linear complexity.

**Strengths:**

1. Clear which–where–what decomposition and a linear-time 3WD decoding mechanism; thoughtful integration of luminance/chrominance branches plus FFT-based cues for degradation localization.
2. Strong empirical coverage: common, compound, and unseen degradations; consistent average PSNR/SSIM gains; tangible downstream benefits on object detection.
3. Solid ablations (latent priors, DM, µ-fusion, 3WD) and efficiency analysis (FLOPs, linear attention vs. SA/CA) that support the design choices.

**Weaknesses:**

1. Many benchmarks are synthetic (e.g., SOTS, Rain100L/CCD, compound datasets). Real-capture datasets (wild rain/snow/low-light, RAW/ISP pipelines) are underrepresented; robustness to sensor noise/ISP and compression artifacts remains unclear.
2. Heavy reliance on PSNR/SSIM; limited perceptual/user-study evidence. Including LPIPS, MUSIQ/NIQE, or human preference tests would better validate perceptual gains and avoid PSNR-overfitting.
3. While prior AIR (PromptIR/ADAIR/DFPIR/DiffUIR) are covered, results against task-specialized SOTA per degradation (e.g., latest LLIE, dehaze, derain, deblur transformers/diffusions) are not uniformly comprehensive on each dataset split.
4. The encoder pretraining mixes single/compound degradations, then is frozen; risk of domain overfitting or latent collapse is only qualitatively addressed. Quantitative analyses of latent separability/stability across seeds and cameras would help.
5. Degradation maps are compelling but lack quantitative localization validation (e.g., overlap with synthetic corruption masks, perturbation or pointing-game metrics). Similarly, how µ-fusion decisions correlate with true degradation semantics is not rigorously measured.
6. Linear 3WD is attractive, but end-to-end latency on 2–8K images, memory usage under tiling, and comparison to windowed attention are not reported.

**Questions:**

1. Do latent priors transfer to real-capture datasets (e.g., LLIE in the wild, real rain/snow) and RAW pipelines? Any results on smartphone benchmarks or cross-sensor generalization without finetuning?
2. How stable are the learned {xℓ, µ} across runs/seeds? Provide cluster separability (t-SNE/UMAP) with degradation labels, NMI/ARI, and center/variance over seeds; any evidence of latent collapse under heavy compound degradations?
3. On synthetic corruptions with known masks, what is the IoU/ROC of DM against ground truth per degradation (rain streaks, haze veil, snow flakes, low-light masks)? Does DM remain calibrated under unseen mixes?
4. Sensitivity to FFT features (remove magnitude/phase), LC branching (luma-only vs chroma-only), FiLM parameterization (γ/β forms), and temperature/β in VAE loss? Please include variance across 3–5 runs.
5. Could you report LPIPS/MUSIQ or a user study on a subset? Any failure cases (e.g., color shifts, ringing, detail hallucination) compared to PromptIR/ADAIR?

---

> ### Author Response · Authors · 2025-11-24
> **Rebuttal Part 1 (C1 - C3)**
>
> ## **R1.1: Many benchmarks are synthetic.**
>
> Thank you for your feedback. To ensure realism and broad applicability, we employ the LSD dataset [1] (the largest single-shot low-light enhancement dataset) collected in uncontrolled, real-world settings across 10+ smartphone sensors. We introduce random noise (instead of fixed patterns) for general denoising evaluation and use the larger RAIN100H [2] dataset for more challenging rain removal scenarios. We further validate our method on real-world datasets, including LSD-U [1] (collected from different sensors and platforms, including social media), real-world denoising with SSID, underwater image enhancement, and out-of-domain medical images. Our experiments span diverse, out-of-distribution sensors and unseen degradation (zero-shot). Notably, our approach achieves significant gains over state-of-the-art methods, improving NIQE by 1.60 and MUSIQ by 4.09 on real-world benchmarks, demonstrating robustness and practical relevance.
>
>
> ## **R1.2: Heavy reliance on PSNR/SSIM; limited perceptual/user-study evidence**
>
> Thank you for your insightful comment. We evaluated our method beyond PSNR/SSIM to ensure perceptual enhancement. In the initial submission, we excluded LPIPS to keep tables concise. We have now added perceptual metrics for SOTA comparison:
>
> *Table1: Performance on unseen real-world restoration tasks using no-reference metrics*
> | Method     | Average (NIQE↓ / MUSIQ↑ / BRISQUE↓) |
> |-----------|--------------------------------------|
> | Input     | 8.92 / 44.42 / 32.58                 |
> | PromptIR  | 9.65 / 43.81 / 40.03                 |
> | DFPIR     | 8.01 / 45.02 / 25.52                 |
> | ADAIR     | 6.85 / 46.10 / 28.45                 |
> | **DAIR (Ours)** | **5.25 / 50.19 / 25.97**     |
> | Improvement | −1.60 / +4.09 / +0.45             |
>
> Our method outperforms SOTA by 0.048 and 0.022 in LPIPS for common and compound restoration, respectively. We will update all tables with LPIPS for consistency. Additionally, we evaluated on real-world unseen tasks with NIQE/MUSIQ/BRISQUE as shown in the first table above.
>
> *Table 2: Performance on six common degradation types*
>
> | Method        | Params (M) | GFLOPs | Avg (PSNR↑ / SSIM↑ / LPIPS↓) |
> |---------------|------------|--------|-------------------------------|
> | Uformer       | 20.63      | 43.86  | 24.34 / 0.8092 / 0.1584       |
> | Restormer     | 26.13      | 141.75 | 23.10 / 0.7718 / 0.1973       |
> | AIRNet        | 8.93       | 301.27 | 21.75 / 0.7583 / 0.1918       |
> | PromptIR      | 35.59      | 158.14 | 25.37 / 0.7977 / 0.1600       |
> | DiffUIR       | 36.26      | 4400.00| 22.32 / 0.7694 / 0.1870       |
> | ADAIR         | 28.78      | 147.18 | 25.35 / 0.8231 / 0.1491       |
> | DFPIR         | 31.07      | 151.07 | 24.46 / 0.7977 / 0.1890       |
> | **DAIR (Ours)** | **18.08** | **45.65** | **28.15 / 0.8829 / 0.1011** |
> | Improvement   | −10.70     | −96.10 | +2.78 / +0.0598 / −0.0480     |
>
> *Table 3: Performance on five compound degradation types*
>
> | Method        | Average (PSNR↑ / SSIM↑ / LPIPS↓) |
> |---------------|----------------------------------|
> | Uformer       | 20.68 / 0.7786 / 0.2007           |
> | Restormer     | 20.79 / 0.7400 / 0.2304           |
> | AIRNet        | 14.10 / 0.6572 / 0.3364           |
> | PromptIR      | 20.28 / 0.7725 / 0.2180           |
> | DiffUIR       | 15.28 / 0.6854 / 0.2823           |
> | ADAIR         | 21.83 / 0.7948 / 0.1769           |
> | DFPIR         | 18.78 / 0.7188 / 0.2656           |
> | **DAIR (Ours)** | **22.61 / 0.8277 / 0.1554**     |
> | Improvement   | +0.78 / +0.0329 / −0.0215         |
>
>
> ## **R1.3: Results against task-specialized SOTA per degradation**
>
> Thank you very much for your insightful feedback. We would like to point out that many task-specific restoration tasks can be expressed using an additive model:
>
> $I_{\mathrm{deg}} = I_{\mathrm{gt}} + \mathrm{Deg}$
>
> where **Deg** denotes the additive degradation (e.g., noise, snow, blur, etc.). Some degradations, such as lowlight conditions, illustrate more complex or non-additive processes and often require nonlinear modeling. By including desnowing and lowlight enhancement, our study covers a wide spectrum of real-world challenges.
>
> We want to emphasize that our primary motivation is to enable all-in-one image restoration in complex scenarios without manual prompting or user guidance. Accordingly, our experiments are designed to evaluate both AIR methods and specialized SOTA baselines under unified, challenging conditions, rather than focusing on individual task-specific settings. While we include challenging task-specific cases, such as desnowing and real-world lowlight enhancement, to provide insight into special scenarios, the manuscript’s primary focus remains on the AIR setting. By benchmarking across 18 degradation types (11 seen, 7 unseen), our evaluation aims to robustly and comprehensively demonstrate generalizability in AIR settings, thereby supporting our motivation.

---

> ### Author Response · Authors · 2025-11-24
> **Rebuttal Part 2 (C4)**
>
> ## **R1.4:  Quantitative analyses of latent separability/stability**
>
> Thank you for your comment. We observed that the latent space of our VAE is well-separated for both seen and unseen degradations. In response to your suggestion, we quantitatively evaluated latent separation at different stages, including experiments without our proposed SupCon weak supervision. For unseen degradations, we considered compound cases (e.g., Low-light + Rain) as well as out-of-distribution tasks such as underwater image enhancement. Our findings are quantified using KNN accuracy and separation ratio, as detailed in Table 4 below:
>
> *Table 4: Clustering Performance in Latent Space — Seen vs. Unseen Classes*
>
> | Model            | Acc↑ (Seen) | Sep↑ (Seen) | Acc↑ (Unseen) | Sep↑ (Unseen) | Acc↑ (Overall) | Sep↑ (Overall) | DB↓  | CH↑     |
> |------------------|-------------|-------------|----------------|----------------|-----------------|-----------------|------|---------|
> | **Steps = 0**        | 0.826       | 3.108       | 0.756          | 1.331          | 0.844           | 2.284           | 4.843 | 458.34  |
> | **Steps = 100k**      | 0.871       | 2.360       | 0.828          | 1.274          | 0.892           | 1.950           | 2.451 | 289.84  |
> | **w/o SupCon**       | 0.864       | 2.006       | 0.787          | 1.323          | 0.874           | 1.788           | 2.690 | 254.48  |
> | **Full Training**    | **0.994**   | **2.911**   | **0.976**      | **1.523**      | **0.991**       | **3.222**       | **1.227** | **549.34** |
>
> Our VAE achieves strong clustering performance in the latent space for both seen and unseen degradation classes. Notably, the full training setup with SupCon supervision yields the highest KNN accuracy (0.994 for seen and 0.976 for unseen) and separation ratio (2.911 for seen and 1.523 for unseen), significantly outperforming the baseline and ablated settings. Global metrics such as Davies-Bouldin (DB) index and Calinski-Harabasz (CH) score further confirm the effectiveness of our approach, with the lowest DB and highest CH values under full training. These results demonstrate that our learned degradation priors enable precise discrimination among degradation types, providing robust guidance for DAIR and improving restoration performance across diverse scenarios.

---

> > ### Author Response · Authors · 2025-11-24
> > **Rebuttal Part 3 (C5)**
> >
> > ## **R1.5: Degradation maps are compelling but lack quantitative localization validation**
> >
> > Following your valuable feedback, we conducted a per-degradation localization analysis to evaluate the performance of each degradation map level. Unfortunately, we could not find any reliable synthesized masks for the specific degradation types in our dataset. To address this, we generated ground truth (GT) degradation maps by computing the pixel-wise difference between the degraded and clean images for each degradation type. We extracted over 3,000 GT degradation maps from our testing set and directly compared them against our predicted degradation maps at each level. The results of this degradation extraction and comparison analysis are summarized in Table 5 below:
> >
> > *Table 5: Quantitative Localization Validation of Degradation Maps*
> >
> > | Metric         | Noise (L3)           | Lowlight (L3)        | Haze (L1)           | Rain (L3)           | Snow (L3)           | Blur (L3)           | Mean ± Std             |
> > |----------------|-----------------------|------------------------|----------------------|----------------------|----------------------|----------------------|-------------------------|
> > | **IoU ↑**      | 0.9975 ± 0.0014       | 0.9643 ± 0.0749        | 0.9593 ± 0.0636      | 0.9103 ± 0.0663      | 0.8157 ± 0.1274      | 0.5211 ± 0.1453      | **0.8614 ± 0.1685**     |
> > | **Dice/F1 ↑**  | 0.9988 ± 0.0007       | 0.9800 ± 0.0502        | 0.9780 ± 0.0381      | 0.9517 ± 0.0387      | 0.8928 ± 0.0820      | 0.6725 ± 0.1330      | **0.9123 ± 0.1182**     |
> >
> > Our degradation maps achieve strong overall performance with a mean IoU of 0.8614 and a Dice score of 0.9123. Notably, noise, lowlight, and haze degradations demonstrate strong localization accuracy with IoU scores exceeding 0.95.
> >
> > In addition to your comment, we also perform an ablation study to illustrate the practicability of our proposed degradation map. We summarized the performance in Table 6 below:
> > *Table 6: Analysis of Degradation Map*
> >
> > | Method        | PSNR↑ / SSIM↑ / LPIPS↓          |
> > |---------------|----------------------------------|
> > | w/o FFT       | 22.80 / 0.7769 / 0.2116          |
> > | **Proposed**  | **26.90 / 0.8565 / 0.1316**      |
> > | *Improvement* | **+4.10 / +0.0796 / −0.0800**     |
> >
> > Our proposed FFT component enables the network to better localize degradation regions by exploiting frequency-domain characteristics.
> >
> >
> > To rigorously address µ-fusion decisions and true degradation semantics, we systematically evaluate how different fusion strategies impact restoration performance. We summarized our results in Table 7 below:
> >
> > *Table: Ablation study of fusion strategies.*
> >
> > | Method        | Beta | Alpha | Factor | PSNR↑ / SSIM↑ / LPIPS↓ |
> > |---------------|------|--------|--------|--------------------------|
> > | w/o MU        | ✘    | ✘      | ✘      | 22.74 / 0.7791 / 0.2135 |
> > | w/o beta      | ✘    | ✔      | 1      | 22.97 / 0.7824 / 0.2147 |
> > | alpha + beta  | ✔    | ✔      | 0      | 17.88 / 0.7272 / 0.2488 |
> > | Fusion 0.1    | ✔    | ✔      | 0.1    | 16.28 / 0.7350 / 0.2345 |
> > | Fusion 0.5    | ✔    | ✔      | 0.5    | 22.98 / 0.7837 / 0.2143 |
> > | **Proposed**  | ✔    | ✔      | 1      | **26.90 / 0.8565 / 0.1316** |
> > | *vs. Best*    |      |        |        | **+3.92 / +0.0728 / −0.0827** |
> >
> > Our full µ-fusion (Factor=1) achieves 26.90 dB PSNR, outperforming the second-best by +3.92 dB. Removing degradation-aware components (beta: 22.97 dB, MU-fusion: 22.74 dB) or using fixed fusion factors (Factor=0: 17.88 dB) causes significant performance drops, demonstrating that learned µ-fusion weights correctly capture degradation characteristics and make semantically meaningful decisions.

---

> > > ### Author Response · Authors · 2025-11-24
> > > **Rebuttal Part 4 (C6 - C9)**
> > >
> > > ## **R1.6: End-to-end latency, memory usage under tiling, and comparison to windowed attention are not reported.**
> > >
> > > Following your valuable suggestions, we have added comprehensive comparisons with window attention, including performance metrics, inference latency, and memory consumption for each attention mechanism. Due to hardware memory constraints, we were unable to evaluate SA and CA at HD/2K resolutions, as their quadratic complexity $O(H^2W^2)$ causes out-of-memory errors. Similarly, 4K resolution experiments exceeded our hardware capacity. We summarize our key findings in Table 8 below:
> > >
> > > *Table 8: Attention mechanism comparison. Best in bold, second best in italics.*
> > >
> > > | Method        | Complexity      | Memory 720p (MB)↓ | FPS 720p↑ | Memory 2K (MB)↓ | FPS 2K↑ | PSNR↑ / SSIM↑ / LPIPS↓ |
> > > |---------------|------------------|-------------------|-----------|------------------|---------|--------------------------|
> > > | CA            | O(H²W²)          | -                 | -         | -                | -       | 17.71 / 0.5765 / *0.5227* |
> > > | SA            | O(H²W²)          | -                 | -         | -                | -       | *19.05* / *0.5444* / 0.5621 |
> > > | WA            | O(M²·HW)         | *4519.90*         | *8.77*    | *10145.04*       | *3.92*  | 23.42 / 0.7944 / 0.2024 |
> > > | **3WD (Ours)** | **O(HW)**        | **2045.94**       | **45.69** | **4573.29**      | **20.45** | **26.90 / 0.8565 / 0.1316** |
> > > | *Improvement* | Linear           | −54.7%            | +5.2×     | −54.9%           | +5.2×   | +3.48 / +0.0621 / −0.0708 |
> > >
> > > Our method achieves linear complexity \(O(HW)\) compared to quadratic alternatives \(O(H^2W^2)\), enabling efficient high-resolution processing. At 2K resolution, 3WD reduces memory by 54.9\% and achieves 5.2$\times$ speedup over window attention while delivering +3.48 dB improvement (26.90 vs. 23.42 dB).
> > >
> > > ## **R1.7: Do latent priors transfer to real-capture datasets**
> > >
> > > Thank you for this question. Our latent prior demonstrates strong real-world transferability:
> > >
> > > ### **Real-captured datasets & cross-sensor generalization (zero-shot)**:
> > >
> > > As detailed in Comment #1 and Comment #2, we validate extensively on real-world data:
> > >
> > > - **LLIE in the wild:** LSD-U (10+ smartphone sensors, social media captures) — NIQE ↑1.60, MUSIQ ↑4.09
> > > - **Real denoising:** SSID (multi-sensor smartphone captures)
> > > - **Out-of-domain:** Underwater enhancement, medical image enhancement
> > > - **Latent analysis:** Separation ratio 1.53 on unseen real captures
> > >
> > > ### **Cross-sensor generalization:**
> > >
> > > LSD-U spans 10+ smartphone sensors in uncontrolled conditions, demonstrating robust cross-sensor generalization without fine-tuning.
> > >
> > >
> > > ### **RAW pipelines:**
> > >
> > > Our current implementation operates in sRGB space. Extending to RAW pipelines is a valuable direction for future work, though our strong out-of-domain performance suggests that the latent prior should transfer effectively.
> > >
> > > Please see Comments #1–3 for comprehensive real-world validation details.
> > >
> > >
> > > ## **R1.8: How stable are the learned $\{x_{\ell}, \mu\}$ across runs/seeds?**
> > >
> > >
> > > We tuned the hyperparameters as outlined in Algorithm 1 (Appendix). With our proposed hyperparameter settings, the learned degradation prior remained stable throughout training. However, when we removed the weak supervision provided by the SupCon loss, we observed that the latent space struggled to separate the different degradation types effectively. Despite this, our design choices prevented any absolute collapse of the latent representations.
> > >
> > > To further investigate, we conducted a detailed analysis of how our training strategies converge over time. We summarized our findings in our response to comment 4. In addition to quantitative evaluation, we also visualized the separation of degradation types in the latent space using t-SNE. We will include these visual results in the revised manuscript to provide a clearer illustration of the latent space separation.
> > >
> > >
> > > ## **R1.9: What is the IoU/ROC of DM against ground truth per degradation**
> > > As per your suggestion, perform per degradation analysis. Please see the response to comment \# 4. Our DM remains calibrated in zero-short unseen degradation. We summarized the performance of DAIR on unseen real-world scenes on comment \# 2.

---

> > > > ### Author Response · Authors · 2025-11-24
> > > > **Rebuttal Part 5 (C10 - C11)**
> > > >
> > > > ## **R1.10: Sensitivity to FFT, LC, FiLM, and temperature/$\beta$ in VAE loss?**
> > > >
> > > > As per your important comments, we added a detailed analysis of the degradation map (with/ without FFT), FiLM formulation, and fusion. Please see the response to comment 5 for more details. Also, we analyse "which" component. Please find the summary of our luma/chroma only and proposed method in Table 9 below:
> > > >
> > > > *Table 9: Ablation study of "which" components*
> > > >
> > > > | Method              | PSNR↑ / SSIM↑ / LPIPS↓            |
> > > > |---------------------|------------------------------------|
> > > > | Luma only           | 23.58 / 0.7996 / 0.1946            |
> > > > | Chroma only         | 23.41 / 0.7927 / 0.2014            |
> > > > | **Proposed (Both)** | **26.90 / 0.8565 / 0.1316**        |
> > > > | *Gain (vs. best single)* | **+3.32 / +0.0569 / −0.0630** |
> > > >
> > > >
> > > > Our proposed latent component (LC) separation enhances the framework’s ability to precisely capture both luminance and chrominance information. This separation validates our reasoning, as it allows the model to effectively capture and utilize both structural and chromatic features in the data.
> > > >
> > > >
> > > > ## **R1.11: Could you report LPIPS/MUSIQ or a user study on a subset?**
> > > >
> > > > In response to your comment, we have included LPIPS metrics in the comparison of common and compound restoration tasks (Table 2/3). For the evaluation of unseen real-world restoration, we have summarized results using NIQE, MUSIQ, and BRISQUE scores (Table 1). For more details, kindly see comment \# 2
> > > >
> > > > We observed that our method may occasionally produce over-enhancement, particularly with challenging unseen underwater degradations, and may introduce artifacts such as color shifts and detail hallucinations in extremely low-light conditions (under 5 lux), especially in homogeneous spatial regions. Compared to PromptIR and ADAIR, our approach demonstrates competitive performance but is still susceptible to these issues in some corner cases.
> > > >
> > > >
> > > >
> > > >
> > > > # **References**:
> > > > [1] Sharif, S. M. A., et al. "Illuminating darkness: Enhancing real-world low-light scenes with smartphone images." arXiv preprint arXiv:2503.06898 (2025).
> > > > [2] Yang, Wenhan, et al. "Joint rain detection and removal from a single image with contextualized deep networks." IEEE transactions on pattern analysis and machine intelligence 42.6 (2019): 1377-1393.

---

> > > > > ### Comment · Reviewer_ivbb · 2025-11-28
> > > > > **Response of Reviewer ivbb**
> > > > >
> > > > > After reading the authors' rebuttal and other reviewers' comments, I decided to maintain my score. The authors have addressed most of my concerns, and I'd like to recommend this paper. The initial score has reflected the quality of this paper.

---

### Author Response · Authors · 2025-12-03
**Rebuttal Summary: Revisions, New Experiments, and Positive Reviewer Feedback**

We thank the reviewers (**ivbb**, **9byp**, **jDsD**, **SuKu**) for their constructive feedback. Overall, all reviewers recognized our novel **'Which-Where-What'** paradigm as a significant step toward prompt-free restoration and highlighted the linear complexity **$\mathcal{O}(HW)$** of our 3WD decoder as a key efficiency advantage.

Below, we summarize the consensus reached before the discussion cut-off and the status of key concerns:

| Reviewer | Primary Concern | Status at Discussion Cut-off |
| :--- | :--- | :--- |
| **ivbb** | Synthetic data,  analysis| ✅ **Explicitly recommended Acceptance** following the rebuttal. |
| **SuKu** | Analysis | ✅ **Most concerns were addressed**; final validation added to paper. |
| **jDsD** | Proof of reasoning, comparison | 🔄 **Addressed in revision**; extensive new experiments provided (Sec 4.5, Appendix A-D ). |
| **9byp** | Unseen datasets | 🔄 **Addressed in revision**; 6 new real-world datasets added (Sec. 4.7, Appendix D). |

To support these resolutions, we significantly expanded the manuscript. We added **31** new experiments, **11** analysis tables, and **4** new figures. Crucially, we benchmarked against **6** new real-world datasets (over **4,000** images) to demonstrate robustness beyond synthetic benchmarks. We detail these specific updates below.

---
**Comment: Lack of Proof for "Reasoning" Modules (jDsD, SuKu, ivbb)**

We validate the "Which-Where-What" paradigm empirically:

**"Which"**: Latent encoding shifts **feature selection** by degradation type (structural vs. chromatic).

**"Where"**: Degradation maps achieve high **IoU and localization** accuracy vs. ground-truth.

**"What"**: Latent fusion captures degradation **semantics**, guiding reconstruction without prompts.

**Action:** We added Section 4.5 to detail this analysis, supported by Table 4, Figure 6, and Table 7 in the main manuscript. We included extensive module-wise ablation studies in Tables 13–15 of the Appendix.

---

**Comment: Latent separability (ivbb), degradation capture (SuKu), SupCon for VAE pretraining (jDsD), apple-to-apple comparison (9byp)**

Latent descriptor pretraining with weak SupCon guidance is one-time. Key points:

- *SupCon role*: Guides corruption separation and training stability; not used at inference.
- *Generalization*: Latent descriptor learns degradation representations progressively, showing **strong separation** for seen and unseen corruptions.

**Action**: Added Table 4 (quantitative analysis) and Figure 8 (Appendix A.2, visual evidence).

---

**Comment: benchmarks are synthetic (ivbb), benchmarking on Real-dataset (SuKu), Unseen Dataset (9byp)**

We evaluated **six real-world out-of-distribution degradations** (**4,000+** images): (1) real denoising, (2) unseen low-light (10+ sensors), (3) underwater enhancement, (4-5) medical enhancement/denoising, and (6) compound degradations (blur + low-light).

**Action:** Table 10 (main paper); Tables 21-22, Figures 11-13 (Appendix D.2-D.3).

---

**Comment: Heavy Reliance on PSNR/SSIM; Limited Perceptual Metrics (ivbb)**

We employ a comprehensive evaluation combining reference-based perceptual metrics (**LPIPS**) and no-reference metrics (**NIQE, MUSIQ, BRISQUE**) for real-world validation.

**Action:** Added LPIPS to Tables 1–3 (main paper) and Tables 12–14, 22 (appendix). Added no-reference evaluations in Table 11 (main paper) and Tables 21–22 (appendix).

---

**Comment: Results Against Task-Specialized Methods (ivbb, jDsD)**

Our primary contribution is a novel reframing of all-in-one restoration (AIR) without explicit prompting. We validated on 11 seen and 8 unseen degradation types, demonstrating versatility on distinct, non-linear challenges (e.g., desnowing, LLIE).

**Action:** Table 2 shows desnowing and low-light results; see Tables 17–18 (appendix) for additional results.

---

**Comment: Latency, Memory Usage, and Comparison to Windowed Attention (ivbb, 9byp)**

Our 3WD is **54.7%** more memory efficient and **5.2×** faster than window attention on 720p and 2K images.

**Action:** Added Table 8 (main paper).

---

**Comment: Standard Experimental Settings (jDsD)**

Our focus is real-world AIR (common and compound settings). We also benchmarked on the standard three-task setting, outperforming SOTA by **+1.58 dB PSNR, +0.031 SSIM**.

**Action:** Table 16 (Appendix C.1).

---

**Comments: Typos, Citation Formatting, Figure Clarity (jDsD, SuKu)**

We revised figures, corrected citations, and updated formatting per reviewer requests.

**Action:** Revised Figures 1–2; added missing citations and corrected formatting to comply with ICLR standards.

---

## **Additional Note**

**Reviewer SuKu** requested further validation of the **"Which" and "What" modules** beyond module-wise analysis. We addressed this with additional theoretical and empirical evidence in **Sections 4.1 and 4.3** (added post-discussion).

---
## **Conclusion**
We believe this work makes a valuable addition to the **ICLR** program.

---

### Meta-Review · Area_Chair_pr9m · 2025-12-13

**Summary:**

Although the paper has appealing motivation and engineering aspects (e.g., reframing all-in-one restoration as latent prior inference and emphasizing an efficient linear-complexity decoder), and reviewers acknowledge the coherence and efficiency value, I recommend Reject. A key contextual factor is that before the system bug occurred, there was limited effective back-and-forth between reviewers and the authors, so the central disagreements were not sufficiently clarified or converged within the review period.

1. Insufficient evidence for the central “which/what” contribution (reads more like narrative than validated mechanism).
The high-confidence reject explicitly argues that the “which–where–what” reasoning framing lacks convincing, targeted evidence and is closer to post-hoc narrative than mechanism-level validation. Moreover, even after rebuttal additions, concerns remain that the new evidence supports “where” more clearly, while “which/what” are still validated mainly via performance gains rather than role-faithful, targeted verification. This directly impacts the paper’s primary novelty claim.

2. Structural concern on protocol standardness and comparison fairness. The high-confidence reject notes that the evaluation setting is not well aligned with standard AIR protocols and requests PromptIR/Perceive-IR style settings for apples-to-apples comparisons. While the authors provide additional results in the rebuttal, the review record does not show clear consensus that the paper’s main conclusions are robust under standard protocols rather than being tied to a customized heterogeneous setting.

3. Tension in the “no prompt / no degradation supervision” narrative. A reviewer highlights that SupCon pretraining uses degradation labels, implying reliance on degradation supervision during training, which conflicts with the paper’s messaging around removing external cues. This key claim requires tighter framing and stronger, explicitly controlled evidence.

I regret to recommend rejection of this submission. Unfortunately, after the system bug occurred, the authors and reviewers did not have a further opportunity to continue the discussion, so I must base my decision on the current review record and evidence available in the manuscript and rebuttal. I encourage the authors to consolidate the additional experiments and analyses introduced in the rebuttal into the main paper, strengthen the mechanism-aligned validation (especially for the “which/what” components), and provide clearer apples-to-apples evaluations under standard AIR protocols. With these improvements, a future resubmission could be substantially stronger.

**Reviewer Concerns:**

R1 (ivbb): The rebuttal adds extensive evidence on (i) broader real-world/OOD evaluation, (ii) perceptual and no-reference metrics beyond PSNR/SSIM (e.g., LPIPS and several NR-IQA metrics), (iii) latent-space separability/analysis, (iv) quantitative validation of the degradation map (e.g., IoU/Dice), and (v) runtime/memory comparisons against window-attention baselines.

R2 (9byp): The rebuttal clarifies the conceptual gap vs. prior non-prompt AIR methods, discusses apples-to-apples training considerations, and adds concrete efficiency evidence (speed/memory tables) plus stronger OOD/real-data validation.

R3 (jDsD): The rebuttal responds to the “non-standard setting” complaint by adding results under a standard 3-task setting, and it promises clearer presentation (updated main figure and degradation-map usage). It also provides additional ablations/quantitative analyses intended to support the “which/where/what” framing.

R4 (SuKu): Based on the author discussion summary, most analysis/validation-related concerns were addressed by the added experiments and clarifications.

High-resolution end-to-end latency/memory evidence is incomplete; some baselines are reported as OOM at higher resolutions and the comparisons are therefore not fully comprehensive. The alignment against the strongest task-specialized SOTA for each degradation type may still be incomplete; the paper emphasizes AIR, but this can remain a credibility concern for some reviewers.

**Reviewer Scores:**

R1 (ivbb): 6 → 6 (no change). R1 explicitly stated they would maintain their score after reading the rebuttal and recommended the paper.

R2 (9byp): 4 → 6 (likely +2). The rebuttal directly addresses R2’s main concerns (conceptual positioning, apples-to-apples discussion, concrete speed/memory evidence, and broader OOD/real-data evaluation), so a modest upward revision is plausible if fully discussed.

R3 (jDsD): 2 → 4 (possibly 4 if convinced by the added standard-setting results and mechanism analyses). The authors added standard 3-task setting results and additional “which/where/what” evidence, but the reviewer’s original concerns about protocol standardness, claim strength, and presentation clarity may still limit how far the score would move.

R4 (SuKu): Small positive shift (e.g., +2). Per the author discussion summary, most analysis-related concerns were addressed with added validation, so a modest bump is plausible if the reviewer had participated fully through the end of discussion.

---

### Decision · Program_Chairs · 2026-01-26

Reject